# Car4Cast: A Dataset and Benchmark for LLM-Based Motion Forecasting and Spatial Reasoning in Autonomous Driving

## Abstract

Recent advances in Large Language Models (LLMs) have shown promise in diverse reasoning tasks, yet their ability to perform structured spatial-temporal prediction remains underexplored. To address this, we introduce Car4Cast, a novel dataset and benchmark that casts 3D motion forecasting in autonomous driving as a spatial reasoning task and testbed, involving structured text generation. Car4Cast provides a comprehensive evaluation suite tailored to the unique challenges of language-based motion prediction, including both classical trajectory accuracy and LLM-specific issues, such as output formatting and hallucinations. Our benchmark also supports an optional visual modality, enabling future exploration of vision-language models in spatial reasoning tasks. Car4Cast is conceived to drive progress toward spatially intelligent language models, highlighting the need and providing data and evaluation tools for new methods and training paradigms that effectively bridge this existing gap.

## 1 Introduction

Large Language Models (LLMs) have shown impressive progress across diverse reasoning tasks, including mathematical, commonsense, and tabular reasoning. A key factor behind these advances is the paradigm of reasoning models, where intermediate steps ("thoughts") enable more structured, human-like logical processes (Xu et al., 2025).

Despite this progress in verbal reasoning tasks, spatial reasoning remains underexplored. Spatial reasoning—the ability to represent, manipulate, and reason about spatial relations—is central to human cognition and critical for real-world applications such as autonomous driving, where perception and planning depend on understanding relative positions, orientations, and interactions among traffic agents (Zhang et al., 2025; Tian et al., 2025).

A major obstacle to improving spatial reasoning capabilities in LLMs is the lack of suitable training and evaluation datasets (Zhang et al., 2025), as current benchmarks provide limited support for structured spatial reasoning, often focusing on qualitative descriptions or vision-centric approaches (Jiang et al., 2024; Ma et al., 2024). This leaves open a key question: *Can reasoning LLMs be leveraged to perform structured spatial reasoning tasks in autonomous driving?*

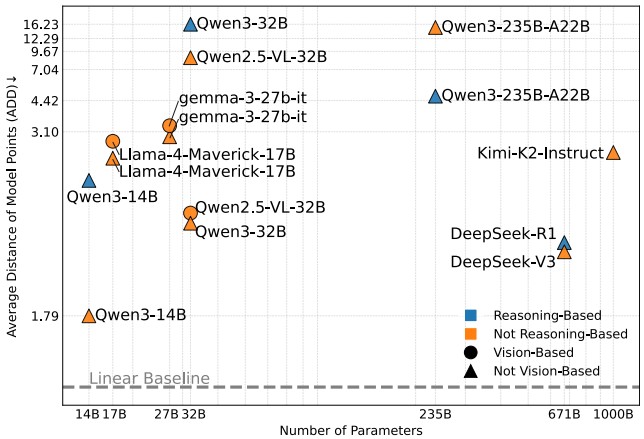

Figure 1: Performance comparison of pre-trained language models on our Car4Cast benchmark in the Average Distance of Model Points (ADD) metric (non-linear axes scale), showing that all tested models underperform a simple linear baseline when averaged over the test set.

We reckon that dedicated datasets and benchmarks are essential for quantifying current LLM spatial reasoning capabilities and guiding future development. To this end, we introduce **Car4Cast**, a dataset and benchmark that reframes the task of motion forecasting (i.e., predicting future trajectories of traffic agents) as a testbed for evaluating spatial reasoning in LLMs. Motion forecasting is particularly suitable to train and evaluate spatial reasoning capabilities because it (i) requires relating numerical data to spatial and physical quantities, (ii) demands spatio-temporal understanding of trajectories and modeling of interactions among agents, (iii) provides clear numerical supervision, enabling quantitative and structured evaluation, and (iv) is an intuitive task for humans yet nontrivial for LLMs, making it highly informative about their inductive biases and reasoning strategies.

Car4Cast consists of nearly 11,000 scenes and over 100,000 trajectories from established 3D autonomous driving datasets, converted into a JSON-like format containing object IDs, past and future positions, rotations, sizes, and semantic descriptors. We choose to present inputs as structured text, as LLMs are well-suited to ingest this modality based on their demonstrated success in other tasks. We further provide a visual modality in the form of top-down LiDAR-based maps, enabling evaluation with Vision-Language Models (VLMs) and supporting multi-modal extensions.

To assess model performance, we provide a tailored evaluation suite. In addition to standard motion forecasting metrics (e.g., displacement error), we incorporate 3D pose estimation metrics to include rotation accuracy, and introduce custom criteria to handle challenges specific to LLM outputs, such as missing entities, hallucinated objects, and formatting inconsistencies.

We further conduct an extensive set of experiments involving a range of pre-trained LLMs and VLMs, covering different model sizes and reasoning paradigms, as well as supervised fine-tuning on our dataset. Our findings reveal brittle LLM behavior, limited gains from model size scaling, and inconsistent benefits from explicit reasoning. VLMs make poor use of visual inputs, and supervised fine-tuning improves output formatting but often degrades numerical accuracy, highlighting a fundamental mismatch between language modeling objectives and structured spatial prediction.

To support future research in spatial reasoning, we will release the Car4Cast dataset, evaluation tools, and finetuning code. Our goal is to provide a rigorous platform for training, evaluating, and improving methods, fostering the development of novel spatially-aware language models.

In summary, our main contributions are:

- A novel benchmark task of 3D motion forecasting as a spatial reasoning challenge for LLMs.
- A dataset of structured text trajectories and optional LiDAR maps from real-world driving scenes.
- A unified evaluation suite with 3D forecasting and LLM-specific structured prediction metrics.
- An empirical analysis revealing limitations of current LLMs and VLMs, underscoring the need for improved methods and training paradigms, while also aiming to inspire the community to develop novel research directions using our dataset, benchmark, and tools.

## 2 RELATED WORK

### 2.1 GENERAL LLM REASONING DATASETS

LLMs have been evaluated on a broad array of verbal, mathematical, and logical reasoning tasks.

**Mathematical reasoning** tests arithmetic and problem-solving skills, with datasets including GSM8K (Cobbe et al., 2021), MATH (Hendrycks et al., 2021), AQuA-RAT (Ling et al., 2017), and SVAMP (Patel et al., 2021). **Multi-hop and deductive reasoning** evaluates compositional and logical inference, with benchmarks including HotpotQA (Yang et al., 2018), EntailmentBank (Dalvi et al., 2022), and MuSiQue (Trivedi et al., 2022). **Tabular reasoning** involves interpreting structured data such as tables, with examples like TabFact (Chen et al., 2020) and WikiTQ (Pasupat & Liang, 2015). **Code and formal reasoning** assesses logical rigor and code understanding, with HumanEval (Chen et al., 2021), MBPP (Austin et al., 2021), and ProofWriter (Tafjord et al., 2021).

### 2.2 SPATIAL/RELATIONAL REASONING IN LANGUAGE MODELS

Spatial and relational reasoning challenge language and multimodal models to interpret object configurations, metric quantities (e.g., size, distance), and spatial relations (e.g., above, next to, inside).

**Diagram Understanding** focuses on extracting structured spatial information from diagrams, charts, or scientific illustrations. Example datasets include AI2D (Kembhavi et al., 2016), DocVQA (Mathew et al., 2021), and InfographicVQA (Mathew et al., 2022). **Path Planning** with language involves interpreting spatial instructions to navigate through physical or simulated spaces. Relevant benchmarks include Room-to-Room (R2R) (Anderson et al., 2018), Touchdown (Chen et al., 2019), ALFRED (Shridhar et al., 2020), and TEACh (Padmakumar et al., 2022).

**Spatial Visual Question Answering (VQA)** extends traditional VQA by emphasizing metric reasoning, such as estimating distances and relative positions. Early benchmarks like CLEVR (Johnson et al., 2017) and GQA (Hudson & Manning, 2019) focused on controlled indoor scenes with simple relational concepts. More recent benchmarks tackle complex environments while incorporating multimodal information. SpatialVLM (Chen et al., 2024) uses 2D segmentation and depth to generate 3D VQA data; VLM-3R (Fan et al., 2025) integrates geometric and camera-view tokens for monocular video inputs; MM-Spatial (Daxberger et al., 2025) leverages depth and multi-view inputs for 3D relational and metric tasks; VSI-Bench (Yang et al., 2025b) provides natural language tasks in indoor environments; Spatial-MLLM (Wu et al., 2025) augments LLMs with 3D foundation model features; SpatialRGPT (Cheng et al., 2024) constructs 3D scene graphs and fuses depth and semantic cues; SpatialLLM (Ma et al., 2025) introduces a 6D-object-orientation-informed VQA dataset and a multi-stage training recipe to incorporate 3D data; NuScenes-SpatialQA (Tian et al., 2025) focuses on metric and situational reasoning in autonomous driving scenes. Similarly to our work, they also exploit annotations from public driving datasets. While aiming to address a similar matter, our work proposes an inherently more abstracted task, focusing on the motion of pre-defined instances rather than the low-level information from camera inputs, and comprising structured numerical predictions.

### 2.3 LANGUAGE-BASED MOTION FORECASTING

Recent work explores the use of LLMs to enhance scene understanding and prediction in motion forecasting. TrajLLM (Lan et al., 2024) integrates a pre-trained LLM with agent-wise descriptors and driving map data to predict future positions. Peng et al. (2025) employ LoRA-based fine-tuning (Hu et al., 2022) for lane-change prediction, while Wang et al. (2025) derive qualitative motion patterns from past trajectories as textual cues combined with road-graph features. Zheng et al. (2024) augment a Motion Transformer with LLM-generated scene descriptions and qualitative action plans. Additionally, Liao et al. (2025) use Chain-of-Thought prompting to decompose prediction tasks and introduce the Highway-Text and Urban-Text datasets with qualitative maneuver descriptions.

### 2.4 MOTION FORECASTING DATASETS

Several large-scale datasets have been developed to support motion forecasting in autonomous driving scenarios. Among the most widely used are ArgoVerse (Chang et al., 2019; Wilson et al., 2021), Waymo Open Dataset (Sun et al., 2020), and ApolloScape (Huang et al., 2020). These datasets provide 2D annotated vehicle trajectories along with high-definition maps and sensor data. While well-suited for benchmarking classical and deep learning-based forecasting models, they are not designed for use with large language models, as they do not provide tools for structured textual outputs or evaluation procedures that address the unique challenges faced by LLMs in this setting.

In contrast, our dataset introduces 3D vehicle trajectories and is specifically designed to evaluate LLMs' spatial reasoning and geometrical coherence, accounting for common failures in LLM-generated structured predictions. Importantly, our dataset does not aim to replace or compete with traditional motion forecasting datasets. Instead, it targets a distinct problem: evaluating and promoting LLMs' understanding of spatial dynamics and temporal consistency in motion prediction.

## 3 THE CAR4CAST DATASET

### 3.1 TASK DEFINITION

We define a trajectory prediction task in which an LLM is required to forecast the future 3D positions and orientations of vehicles in the form of 3D bounding boxes, over a fixed prediction horizon of $T_f = 8$ timesteps, given historical data from the past $T_h = 8$ timesteps. The choice of the full horizon $T = T_h + T_f = 16$ is determined by the minimum scene duration among source datasets.

Each vehicle at any timestep $t \in \{1, \ldots, T\}$, is represented as a 3D bounding box with:

- A **centroid translation** $\mathbf{t}^{(t)} \in \mathbb{R}^3$, representing the 3D position of the box centroid in a global reference frame.
- A set of **Euler angles** $\boldsymbol{\theta}^{(t)} = (\theta_{\text{roll}}^{(t)}, \theta_{\text{pitch}}^{(t)}, \theta_{\text{yaw}}^{(t)}) \in \mathbb{R}^3$, defining the global orientation of the box as a rotation applied in the ZYX (yaw-pitch-roll) order.

Given a history of bounding box motion over $T_h = 8$ timesteps, $\{(\mathbf{t}^{(t)}, \boldsymbol{\theta}^{(t)})\}_{t=1}^{T_h}$, the LLM is tasked with predicting the future centroid translations and Euler angles, $\{(\hat{\mathbf{t}}^{(t)}, \hat{\boldsymbol{\theta}}^{(t)})\}_{t=T_h+1}^{T}$.

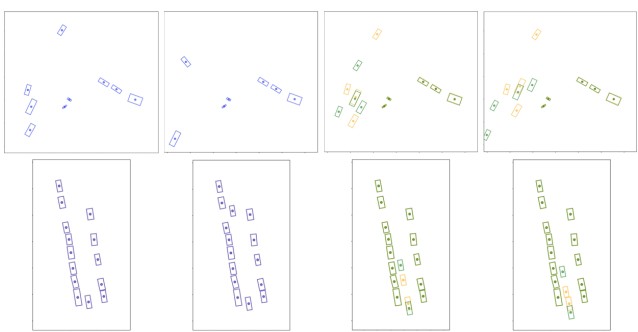

Figure 2: Top-down visualizations of historical (blue), ground-truth future (green), and predicted future (yellow) trajectories. Animated visualizations are provided in the video supplementary material.

## 3.2 DATA FORMAT

Data are stored as structured text in a JSON-like dictionary format (an example is in Appendix A). Each entry corresponds to a scene, which contains multiple vehicle instances. Every scene is identified by a unique `scene_id` and spans two identically-formatted files: a *history file* providing past motion data over $T_h = 8$ timesteps, and a *future file* providing the ground-truth future over $T_f = 8$ timesteps. Each file contains per-instance motion data including the following fields:

- `instance_id`: A unique identifier for each vehicle.
- `timestep`: A list of timesteps, one per observation.
- `translation`: A list of 3D global positions (in meters) in the format $[x, y, z]$ for each timestep.
- `rotation`: A list of 3D Euler angles (in radians) in the format [roll, pitch, yaw] for each timestep. Roll and yaw are wrapped to $[-\pi, \pi]$, while pitch is clamped to $[-\pi/2, \pi/2]$.
- `size`: The physical dimensions (in meters) of the vehicle in the format [length, width, height].
- `attribute_label`: Semantic labels (e.g., "Car", "Truck").

In addition to structured motion data, each scene includes a visual modality, in the form of a top-down map (Figure 3) constructed from aggregated LiDAR point clouds, with units aligned to the metric space used in the textual input. Points are color-coded based on height (i.e., their $z$-coordinate), providing visual cues not only about road topology and drivable area, but also about the semantic composition of the environment (e.g., curbs, buildings, vegetation). This modality is designed to support experiments with vision-language models (VLMs) and to encourage research into multi-modal spatial reasoning.

## 3.3 DATA PROCESSING PIPELINE

Our dataset is built by extracting and reformatting annotations from established 3D autonomous driving datasets: nuScenes (Caesar et al., 2020), KITTI (Geiger et al., 2012), PandaSet (Xiao et al., 2021), Argoverse 2 (Wilson et al., 2021), and CADC (Pitropov et al., 2020). These datasets were selected for their open licenses and support for 3D object detection with persistent instance IDs, enabling the construction of temporally consistent trajectories.

We developed a processing pipeline to standardize coordinate frames, sampling rates, and scene composition. Bounding box annotations were transformed into a global reference frame

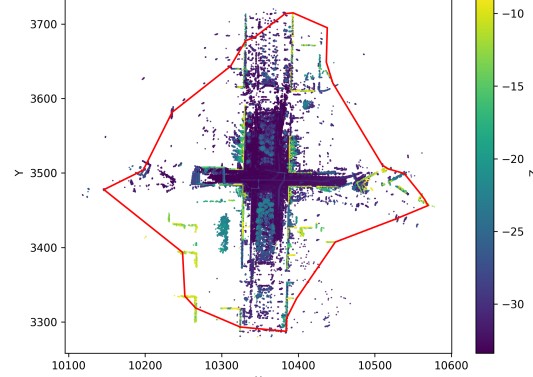

Figure 3: Visual input: top-down LiDAR point-cloud (color-coded by height) and concave hull (red—not part of the model's input).

(defined as the LiDAR frame at the first scene frame) when source annotations were relative (e.g., ego-centric or camera-based), using extrinsic calibration and, when needed, GPS/IMU data.

To unify annotation frequency, all sequences were resampled (or linearly interpolated) at 2 Hz, the lowest native rate among source datasets. Trajectories were segmented into fixed-length sequences of 8 historical and 8 future timesteps. To balance object distributions across scenes and mitigate scene complexity imbalance, we applied spatial clustering based on each object's average position across its 16-timestep span. The number of clusters was chosen such that each sub-scene contains, on average, 10 instances. The resulting $\sim$11,000 sequences were split into train, validation, and test sets in an 80%-10%-10% ratio.

For the visual modality, we generated top-down maps by aggregating LiDAR point clouds across scene frames. Points were transformed to the global frame using calibration and GPS/IMU data when needed, and the overall 3D cloud was projected onto the XY plane to create a bird's-eye view.

# 4 EVALUATION SUITE

To rigorously evaluate LLM-based motion forecasting in Car4Cast, we introduce a comprehensive suite tailored to the unique challenges of structured text-based prediction. Our metrics assess both motion forecasting performance (inspired by classical datasets introduced in 2.4), and issues specific to LLM outputs, such as formatting errors, hallucinated objects, and physical plausibility violations.

## 4.1 EVALUATION METRICS

**Average Distance of Model Points (ADD)**    The primary metric is the **Average Distance of Model Points (ADD)** (Hinterstoisser et al., 2013), a typical evaluation criterion in 3D pose estimation, which jointly captures errors in predicted translation and orientation.
For each timestep $t \in \{T_h + 1, \ldots, T\}$, let:

- $\mathbf{R}^{(t)}$ and $\hat{\mathbf{R}}^{(t)}$ be the ground-truth and predicted rotation matrices from the respective Euler angles,
- $\{\mathbf{x}_i\}_{i=1}^{8}$ be the 3D corners of the bounding box in a local, object-centered reference frame.

The overall ADD over the future prediction window $[T_h + 1, \ldots, T]$ is:

$$\text{ADD} = \frac{1}{T_f} \sum_{t=T_h+1}^{T} \text{ADD}^{(t)}, \quad \text{where} \quad \text{ADD}^{(t)} = \frac{1}{8} \sum_{i=1}^{8} \left\| \mathbf{R}^{(t)}\mathbf{x}_i + \mathbf{t}^{(t)} - \left( \hat{\mathbf{R}}^{(t)}\mathbf{x}_i + \hat{\mathbf{t}}^{(t)} \right) \right\| \quad (1)$$

**Instance Precision, Recall, and F1 Score**    Let $F$ be the set of ground-truth future agents, and $\hat{F}$ the set of predicted future agents, with matching performed via their instance IDs.

$$P_{\text{instance}} = \frac{|\hat{F} \cap F|}{|\hat{F}|} \quad (2) \quad R_{\text{instance}} = \frac{|\hat{F} \cap F|}{|F|} \quad (3) \quad F1_{\text{instance}} = \frac{2 \cdot P_{\text{instance}} \cdot R_{\text{instance}}}{P_{\text{instance}} + R_{\text{instance}}} \quad (4)$$

Low precision indicates many hallucinated (false positive) agents, while low recall reflects many missed (false negative) agents. The F1 score, defined as the harmonic mean of Precision and Recall, combines both metrics into a single criterion. These metrics are especially relevant for LLMs, which may hallucinate or omit entities due to their generative, token-by-token output mechanism.

**Formatting Accuracy (ACC$_f$)**    Let $\hat{F}_{\text{format}} \subseteq \hat{F}$ be the subset of predicted agents that are correctly formatted and can be parsed without errors. Unreadable or malformed instances are repaired using linear interpolation when possible, or assumed to remain static at their last historical state (the detailed policy for malformed outputs is explained in Appendix E).

$$\text{ACC}_{\text{f}} = \frac{|\hat{F}_{\text{format}}|}{|\hat{F}|} \quad (5)$$

Formatting accuracy is a key metric for LLM-based structured prediction tasks, as it reflects the model's ability to generate syntactically correct and structurally valid output. It also distinguishes our benchmark, tailored for language models, from conventional motion forecasting datasets.

**Collision Rate (CR)**    Proportion of predicted agents that collide with another agent at any timestep, where a collision is any nonzero 3D Intersection-over-Union (IoU) between two predicted bounding boxes. This metric evaluates whether models respect spatial interactions and physical constraints.

**Out of Map Rate (OMR)**    OMR measures how often predicted vehicles leave the drivable area, defined using an *alpha shape* (Edelsbrunner et al., 1983) ($\alpha = 0.01$) over the top-down LiDAR point cloud. Alpha shapes are a generalization of the convex hull allowing concavities, with the $\alpha$ parameter controlling the level of detail. A vehicle is considered out-of-map if any predicted position lies outside this boundary and the OMR is reported as the fraction of such vehicles. The low positive alpha value yields a generous boundary (Figure 3), so OMR mainly penalizes clearly implausible trajectories, making it a useful measure of whether VLMs capture scene constraints from the map.

**Average and Final Displacement Error (ADE/FDE)**    The ADE is the average Euclidean distance between predicted and ground-truth centroids across all future timesteps, while the FDE is defined only at the final timestep. They reflect how closely predicted positions track the true path.

**Rotation Error (RE)**    For each Euler angle $\theta^{(t)} \in (\theta_{\text{roll}}^{(t)}, \theta_{\text{pitch}}^{(t)}, \theta_{\text{yaw}}^{(t)})$, define the angular difference:

$$\delta(\theta^{(t)}) = \cos^{-1}\left(\cos\theta^{(t)} \cdot \cos\hat{\theta}^{(t)} + \sin\theta^{(t)} \cdot \sin\hat{\theta}^{(t)}\right) \tag{6}$$

This quantity measures the angular error between predicted and ground truth orientations, accounting for the periodic definition of angles. The final rotation error is computed by averaging the angular difference over all Euler angles and all timesteps for each agent, and then across agents. The RE quantifies how well the model aligns the predicted vehicle orientation to the ground truth.

Mathematical definitions of all metrics are provided in Appendix D.

## 4.2 Motion Categories

Since real-world driving scenes in our dataset are typically dominated by static vehicles (Figure 4), we group agents into three categories for performance stratification, following Peri et al. (2022).

- **Static**: Agents that remain stationary during the forecast window, with a tolerance of 1 meter.
- **Linear**: Agents whose full 16-step trajectory can be approximated by a constant-velocity model (based on first and last positions), with deviations below their average per-timestep displacement.
- **Nonlinear**: Agents exhibiting more complex motion, not belonging to the previous categories.

This categorization is particularly useful to break down the performance of LLMs, which often assume constant-velocity motion (as discussed in 6.1.2 and Appendix F).

## 5 Dataset Statistics

Our dataset comprises a total of **11,985 scenes**, distributed across different subsets as follows: 9,583 in the training set, 1,189 in the validation set, and 1,213 in the test set.

By design (3.3), each scene has a duration of 16 timesteps and contains, on average, 10 instances. In total, the dataset includes **102,510 annotated instances**, categorized as:

- **Static instances**: 80,772 (**78.8%**)
- **Linear instances**: 14,890 (**14.5%**)
- **Nonlinear instances**: 6,848 (**6.7%**).

Using the Qwen3 (Yang et al., 2025a) tokenizer under the ChatML (OpenAI, 2023) format (more details in Appendix B), the dataset exhibits the following token statistics per scene:

- **Minimum**: 1,504 tokens
- **Maximum**: 39,848 tokens
- **Average**: 10,224 tokens per scene.

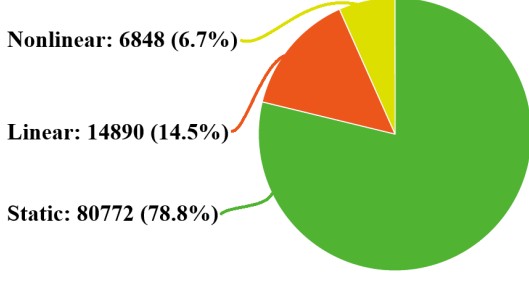

Figure 4: Breakdown of trajectories in the full dataset by motion category.

## 6 EXPERIMENTS

In this section, we evaluate whether current large language models and their reasoning paradigms are suitable for structured 3D motion forecasting. Specifically, we test their ability to process spatiotemporal input, predict future positions and orientations, and generate valid structured outputs.

### 6.1 PRE-TRAINED LLMs

**Models**   We evaluate a range of open-weight pre-trained LLMs, spanning diverse model families, input modalities (both text-only and vision-language), and parameter scales (from 14B to 1000B parameters). The models include:

- DeepSeek: V3 (DeepSeek-AI et al., 2025b) and R1 (DeepSeek-AI et al., 2025a)
- Qwen3 (Yang et al., 2025a): 14B, 32B, and 235B-A22B
- Kimi-K2-Instruct (Kimi-Team et al., 2025)
- Qwen2.5-VL-32B-Instruct (Bai et al., 2025)
- LLaMA-4-Maverick-17B-128E-Instruct-FP8 (MetaAI, 2025)
- Gemma-3-27B-it (Gemma-Team et al., 2025)

Each model is prompted with a task description explaining the forecasting objective and the expected input/output format. The full prompt and inference parameters are provided in Appendices A and B.

**Baseline**   We compare LLM predictions to a simple linear extrapolation baseline, which fits a least-squares linear model over the 8-timestep history for both positions and Euler angles, then extrapolates over the forecast horizon.

**Evaluation**   We compute both **mean** and **median** metrics across all test scenes. Median values are used to mitigate the influence of outliers in distance-based metrics.

#### 6.1.1 OVERALL FORECASTING PERFORMANCE

Across all evaluated models, we observe that:

- All LLMs underperform the linear baseline when evaluated using **mean ADD** (Figure 1).
- Most models outperform the baseline when evaluated using **median ADD** (Figure 5).

This discrepancy suggests that, while LLMs may outperform a trivial baseline in many individual scenes, they also exhibit failure modes that result in large errors, highlighting a lack of robustness. The performance gap between models of different sizes remains narrow, and models in the 670B–1000B parameter range do not demonstrate consistent improvements over smaller counterparts and, in many cases, perform worse. This lack of improvement with scale suggests that the limitations are not simply due to insufficient model capacity or expressivity. Instead, it points to a deeper and more structural deficiency: current training setups do not endow LLMs with the foundational capabilities required for spatial

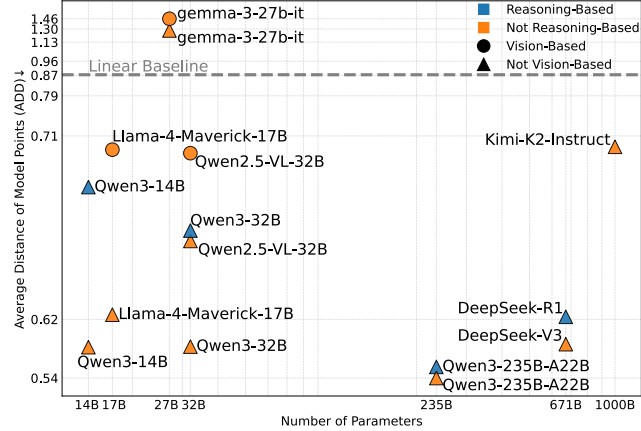

Figure 5: Comparison of pre-trained models in the ADD metric (*overall* motion category), median over the test set (non-linear axes scale).

reasoning. This deficiency likely arises from the absence of appropriate spatially informed training data (a gap that we aim to narrow by introducing and releasing Car4Cast), as well as from objective functions that are unsuitable to the demands of such spatial tasks.

### 6.1.2 EFFECT OF REASONING

To isolate the effect of "thinking" (i.e., explicit step-by-step reasoning) on spatial prediction, we compare in Table 1 models in the Qwen3 family, toggled between "thinking" and "non-thinking" inference modes, as well as DeepSeek-R1, which is reinforcement-learning fine-tuned (i.e., in thinking mode), versus DeepSeek-V3, which does not perform reasoning.

| Model | #Params | ADD ↓ (overall) | ADD ↓ (linear) | ACC$_f$ ↑ | F1$_{instance}$ ↑ |
|---|---|---|---|---|---|
| Qwen3 (NR) | 14B | **0.594** | **2.509** | **0.995** | 0.916 |
| Qwen3 (R) | 14B | 0.666 | 2.982 | 0.977 | **0.924** |
| Qwen3 (NR) | 32B | **0.595** | 2.186 | **0.942** | **0.927** |
| Qwen3 (R) | 32B | 0.653 | 2.576 | 0.918 | 0.925 |
| Qwen3 (NR) | 235B | **0.537** | 1.950 | 0.990 | 0.917 |
| Qwen3 (R) | 235B | 0.562 | **1.926** | **0.996** | **0.927** |
| DeepSeek-V3 | 671B | **0.598** | 2.463 | 0.999 | 0.935 |
| DeepSeek-R1 | 671B | 0.623 | **2.075** | 0.999 | **0.936** |

Table 1: Comparison between reasoning (R) and non-reasoning (NR) models. ACC$_f$ and F1$_{instance}$ are intended as the mean in the *overall* category, ADD is the median.

**ADD (Median)** When evaluated on all instances, "thinking" models perform slightly worse than their "non-thinking" counterparts in terms of median ADD. An exception occurs in the *linear* motion category, where "thinking" models tend to perform better in the large parameter scale ($\geq$ 235B). This aligns with qualitative analysis of reasoning outputs (as shown in Appendix F), where models often explicitly declare the use of a linear motion assumption, resulting in better performance when the agent's motion is in fact linear.

**Formatting and Instance Metrics** When comparing metrics related to the output's structural quality, we observe that **Formatting Accuracy** (ACC$_f$) improves with reasoning in larger models ($\geq$ 235B) and **Instance F1 scores** also tend to be higher in thinking mode, or at least very close.

This likely stems from the fact that reasoning models tend to enumerate instances one by one during the thinking phase, leading to better object persistence, fewer hallucinations, and more syntactically valid outputs—as can be demonstrated by inspection of the thinking process (Appendix F).

### 6.1.3 EFFECT OF VISUAL MODALITY

To evaluate the benefit of map information, we compare vision-language models (i.e., Qwen2.5-VL, LLaMa4, and Gemma-3) with and without access to the top-down map as an image input (Table 2).

| Model | #Params | ADD ↓ | OMR ↓ | ACC$_f$ ↑ |
|---|---|---|---|---|
| Llama-4-Maverick (NV) | 17B | **0.624** | **0.0710** | 0.943 |
| Llama-4-Maverick (V) | 17B | 0.690 | 0.0711 | **0.953** |
| gemma-3 (NV) | 27B | **1.273** | **0.0655** | 1.0 |
| gemma-3 (V) | 27B | 1.464 | 0.0678 | 1.0 |
| Qwen2.5-VL (NV) | 32B | **0.650** | 0.0667 | 0.830 |
| Qwen2.5-VL (V) | 32B | 0.687 | **0.0665** | **0.897** |

Table 2: Comparison between visual (V) and non-visual (NV) models. ACC$_f$ and OMR are intended as the mean in the *overall* category, while ADD is the median.

**ADD and OMR** Vision-enabled models consistently perform worse than their non-vision counterparts in terms of median ADD. Differences in the Out-of-Map Rate (OMR) are minimal, but interestingly, non-vision variants perform slightly better. This suggests the observation that current VLMs do not effectively utilize visual map inputs in this structured forecasting task.

**Formatting** Vision-capable models tend to exhibit better (or equivalent) Formatting Accuracy. This may be a consequence of VQA-style training, which often involves structured answers in JSON-like syntax (e.g., object detection, scene graph extraction) (Bai et al., 2025).

### 6.2 SUPERVISED FINE-TUNING

To assess whether the typical paradigm of task-specific fine-tuning improves language models' performance in spatial reasoning tasks like motion forecasting, we finetune Qwen3-4B using standard Supervised Fine-Tuning (SFT) on the Car4Cast training dataset in the ChatML format (OpenAI, 2023). The detailed training and inference setups are shown in Appendix B.

**Observations** As shown in Table 3, SFT yields notable improvements in formatting-related metrics, e.g., Formatting Accuracy and Instance Precision and Recall. However, it causes degraded performance on numerical distance-based metrics (ADD, ADE, FDE). We argue that this highlights a

|  | ADD ↓ | FDE ↓ | $ACC_f$ ↑ | $P_{instance}$ ↑ | $R_{instance}$ ↑ |
|---|---|---|---|---|---|
| Pre-trained | **0.688** | **0.906** | 0.943 | 0.862 | 0.974 |
| Fine-tuned | 0.956 | 1.254 | **0.995** | **0.875** | **0.989** |

Table 3: Comparison of pretrained and finetuned Qwen3-4B on the test set. ADD and FDE are median, others are mean.

fundamental limitation of standard SFT for structured numerical tasks: the cross-entropy objective optimizes for exact token matching, not numerical accuracy. It does not penalize outputs that are semantically or geometrically close to the ground truth unless they match exactly at the token level. As such, it cannot capture the geometrical or physical relationships required for accurate spatial forecasting. Furthermore, all numeric tokens are treated equally, preventing the model from prioritizing digits that more significantly affect the numerical value (e.g., mistaking 5.5 for 5.6 is treated the same as mistaking 5.5 for 8.5).

### 6.3 SUMMARY OF INSIGHTS

Our findings highlight several insights:

- Pre-trained LLMs can outperform a simple linear baseline in the median case, but still fail on many scenes, leading to worse mean performance.
- Model size alone does not correlate with forecasting accuracy, suggesting that the required capabilities are not a matter of scale but rather of training data and objective.
- "Thinking" improves structural consistency but generally does not yield better geometric accuracy, except in cases where the model's reasoning assumptions match the true motion (e.g., linear).
- VLMs underperform their text-only counterparts when given map images, indicating a lack of visual-spatial grounding in this format.
- Supervised fine-tuning improves formatting consistency but fails to improve numerical accuracy, due to the limitations of token-level objectives.

These results suggest that existing LLMs lack the necessary inductive biases and training objectives to reliably solve structured spatial reasoning tasks like motion forecasting. The full experimental results are reported in Appendix G. Car4Cast is conceived to mitigate this gap at the data level, as well as to encourage the development of more suitable training strategies.

## 7 CONCLUSION

We introduced **Car4Cast**, a dataset and benchmark designed to train and evaluate the spatial reasoning and structured prediction capabilities of LLMs and VLMs in the context of 3D motion forecasting. Car4Cast frames forecasting as a structured generation task over a unified, ingestible textual format, optionally grounded in visual scene context. To quantitatively measure future developments, we proposed an inclusive evaluation suite, comprising both classical forecasting metrics and LLM-specific criteria for formatting accuracy, hallucinated instances, and physical plausibility violations.

Our experimental results, complementary to the dataset and benchmark itself, suggest that while LLMs offer a promising interface for motion forecasting, current training paradigms are inadequate for structured, spatially-informed 3D reasoning. These findings highlight how Car4Cast can serve as a valuable resource, providing data and tools to facilitate the development and evaluation of models for structured 3D spatial reasoning, likely requiring geometry-aware training objectives and schemes (e.g., reinforcement learning-based fine-tuning), more suitable supervision signals, or architectures designed specifically for numerical and physical reasoning in structured domains. In this way, Car4Cast aims to catalyze future research, offering a concrete platform to bridge the existing gap in spatial reasoning for language models.

## REPRODUCIBILITY STATEMENT

We have taken several steps to ensure that our results can be reliably reproduced:

- **Source datasets:** All source datasets used to build Car4Cast are open-source and listed in 3.3.

- **Data processing:** The complete data processing pipeline is described in 3.3. The corresponding source code is provided and documented in the supplementary material.

- **Data sample:** A randomly sampled subset (complying with attachment size limitations) of the dataset is included in the supplementary material. We commit to releasing the full dataset after the review process.

- **Evaluation metrics:** All evaluation metrics are introduced in 4 and defined mathematically in Appendix D. The full evaluation code, including a repair policy for handling malformed outputs explained in Appendix E, is provided and documented in the supplementary material.

- **Forecasting experiments:** The code used to run forecasting experiments is included and documented in the supplementary material.

- **Finetuning experiments:** Similarly, the code used for finetuning experiments is provided and fully documented.

- **Models:** A detailed list of the models used, including public model IDs, is reported in Appendix C.

- **Experimental setups:** The specific setups for fine-tuning and inference are described in Appendix B, with the prompts in Appendix A, and included in the respective code, providing all necessary configuration details.

- **Results:** Complete experimental results are reported in Appendix D.

These resources collectively provide all the information necessary for other researchers to reproduce our findings and build upon this work.

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

# APPENDIX

## A FULL PROMPTS

Listing 1 shows the full prompt for the 3D forecasting task in the non-visual modality.

```
I have time-series data for several vehicle instances, and you
have to forecast the future positions (x, y, z translations) and (
roll, pitch, yaw) angles of these vehicles over the next N
timesteps. The data is structured in a format I will provide. The
output you need to directly provide should be the forecasted
positions and angles in the same format.

Each instance has the following data:

Translations: x, y, z components.
Rotations: roll, pitch, yaw angles.

Data is organized by individual instances, where each instance has
 the following:

instance_id: A unique identifier for each vehicle.
timestep: A list of timesteps for the instance.
translation: A list of [x, y, z] translation coordinates for each
timestep.
rotation: A list of [roll, pitch, yaw] angles for each timestep.
size: The dimensions of the vehicle (not needed for forecasting,
but part of the data).
attribute_label: A list of labels for each instance (not needed
for forecasting, but part of the data).
Example data for instance with ID '4a3dda16-36b3-47dd-9fa4-
b5a220fa0a53':

```json
"4a3dda16-36b3-47dd-9fa4-b5a220fa0a53": {
        "timestep": [
            1,
            2,
            3,
            4
        ],
        "translation": [
            [
                -44.907,
                55.428,
                0.226
            ],
            [
                -39.867,
                50.5,
                0.226
            ],
            [
                -35.738,
                46.458,
                0.232
            ],
            [
```

```
            -31.493,
            42.259,
            0.281
        ]
    ],
    "rotation": [
        [
            0.0,
            0.0,
            -2.3449651189732195
        ],
        [
            0.0,
            0.0,
            -2.3449651189732195
        ],
        [
            0.0,
            0.0,
            -2.3449651189732195
        ],
        [
            0.0,
            0.0,
            -2.3449651189732195
        ]
    ],
    "size": [
        [
            1.623,
            1.913,
            4.441
        ],
        [
            1.623,
            1.913,
            4.441
        ],
        [
            1.623,
            1.913,
            4.441
        ],
        [
            1.623,
            1.913,
            4.441
        ]
    ],
    "attribute_label": [
        "Car",
        "Car",
        "Car",
        "Car"
    ]
}
```

Your Task:

```
972   Forecast the positions: Predict the future Translations (x, y, z)
973   for each vehicle instance at timesteps 8 to 15 with a precision of
974    3 after-comma digits.
975   Forecast the rotations: Predict the future Rotations (roll, pitch,
976    yaw) for each vehicle instance at timesteps 8 to 15 with a
977   precision of 3 after-comma digits.
978   Include the timesteps: You need to include in the final JSON the
979   list of future timesteps at which each translation and rotation
980   occur.
981   You have to employ:
982   Spatial reasoning: Consider the relative positions of all cars at
983   each timestep. Understand and reason about the layout of the scene
984   .
985   Try to figure out how each car is moving (i.e., its direction) as
986   well as how different cars are positioned with respect to one
987   another.
988   Reason about the potential behavior of cars: try to figure out is
989   one is overtaking, or stopping, etc.
990   If any car is in close proximity to another, account for potential
991    interactions or changes in behavior.
992   Temporal reasoning: The trajectory of each car evolves over time.
993   Make sure to maintain continuity and predict the movement in a
994   plausible way given the past trajectory data.
995   Assumptions: You may assume that the cars are following continuous
996    trajectories, and external factors (e.g., traffic lights, road
997   conditions) are not considered in this task.
998
999   You may want to employ some basic geometrical/physical reasoning,
1000  i.e., consider the evolution of the velocity vector over time
1001  rather than the pure x, y, z and rotation values.
1002  We are assuming that cars can't slide or drift, thus the angle of
1003  the velocity vector is exactly the same as the yaw angle.
1004
1005  Ensure the forecasted trajectories are smooth and consistent with
1006  the historical data.
1007  Pay attention to the interactions between cars, such as if they
1008  are close to each other. You may assume basic rules of motion (e.g
1009  ., cars don't teleport or move too abruptly).
1010  Your direct output should be a full JSON file containing the
1011  forecasts only, following the same format as the input. Do not
1012  include the 'size' and 'attribute_label' fields, as they are
1013  constant by nature.
1014
1015  Make sure the JSON content is introduced by "```json" and closed
1016  by "```".
1017
1018  Do not provide code in Python or any other programming language, I
1019   need you to provide the forecast directly.
1020  Please do not simplify the JSON file, even if it contains
1021  repetitive patterns, I need a full file ready to be copy-pasted.
```

Listing 1: Full LLM prompt for 3D motion forecasting in the non-visual modality.

For the VLM modality, the content of Listing 2 is added to the previous prompt after the example JSON, while the content of Listing 3 is added between the "Temporal Reasoning" and the "Assumptions" points.

```
In addition to the JSON file, I will attach an image. This image
is a map representing the top-down view a point cloud collected in
```

```
 the same driving scene, with units representing coordinates in
the same reference frame as the JSON fike. The x and y axes span
the street plane, and their coordinates can be seen on the labeled
 image axes, while z (not visible in this plot) is perpendicular
to the street. Points are color-coded according to their height,
with darker colors (purple) representing lower points and lighter
colors (yellow) representing higher points. White is the plot
background color, so anything in the white region does not belong
to the drivable area. Anything in the colored region (any color
except white) represents the drivable area.
```

Listing 2: Addition to the base prompt for the VLM modality (1/2).

```
Reason about the map: Use the map to guide your forecast, and, in
particular, to ensure the forecasted trajectories stay within the
drivable area.
```

Listing 3: Addition to the base prompt for the VLM modality (2/2).

## B  TRAINING AND INFERENCE SETUPS

### B.1  PRE-TRAINED LLMS

Inference parameters are kept fixed across models: `temperature=0`, `top_k=0`, `top_p=1`, `presence_penalty=0`, `frequency_penalty=0`, and `max_sequence_length` set to the model's limit. A single output is generated per scene ($n = 1$). The use of deterministic decoding (i.e., temperature set to zero) ensures that inference results are fully reproducible without the need to specify a random seed.

### B.2  SUPERVISED FINE-TUNING

**Training Setup**   Each training example is structured as:

- `system`: the full prompt in Listing 1
- `user`: the historical motion dictionary, serialized into structured text
- `assistant`: the ground-truth future motion dictionary in the same textual format (omitted during testing).

We perform LoRA fine-tuning with:

- rank=128, target modules:  [q_proj, k_proj, v_proj, o_proj, gate_proj, up_proj, down_proj]
- `global_batch_size=8`, `num_epochs=2`, `learning_rate=2e-5`, cosine schedule
- `max_sequence_length=40960`

**Inference Parameters**   Following the recommended Qwen3 best practices (Qwen-Team, 2025), we set the following parameters at test-time: `max_new_tokens=32768`, `temperature=0.7`, `top_p=0.8`, `top_k=20`, and `batch_size=1`. Due to the non-deterministic nature of sampling with these parameters, we fixed the random seed to 42 to ensure reproducibility of inference results across runs.

## C  DETAILED LIST OF TESTED MODELS

All "pre-trained" labeled experiments were conducted using publicly available models with open-access checkpoints. For each model, we specify the exact identifier corresponding to its published version to ensure full reproducibility.

The following models were used:

- deepseek-ai/DeepSeek-V3

- `deepseek-ai/DeepSeek-R1`
- `Qwen/Qwen3-14B`
- `Qwen/Qwen3-32B`
- `Qwen/Qwen3-235B-A22B`
- `moonshotai/Kimi-K2-Instruct`
- `Qwen/Qwen2.5-VL-32B-Instruct`
- `meta-llama/Llama-4-Maverick-17B-128E-Instruct-FP8`
- `google/gemma-3-27b-it`

All models were accessed through an inference backend that loads the models directly from their official public checkpoints. Tokenizers and generation logic were used exactly as specified in the respective model repositories.

The fine-tuning experiment was carried out using `unsloth/Qwen3-4B-unsloth-bnb-4bit`.

## D    DETAILED DEFINITION OF EVALUATION METRICS

We provide formal definitions of all evaluation metrics used in the Car4Cast benchmark, expanding on the descriptions in the main paper. Metrics already defined in the main paper (ADD, Instance Precision/Recall/F1 Score, Formatting Accuracy) are not redefined here.

### D.1    MISS RATE (MR@P)

Miss Rate at threshold $p$ is defined as the proportion of agents for which ADD exceeds p% of their ground-truth trajectory length, with a minimum threshold of 1 meter. This is measured over the prediction window $t \in \{T_h + 1, \ldots, T\}$:

$$\text{MR@p} = \frac{1}{N} \sum_{n=1}^{N} \mathbf{1}\left[\text{ADD}^n > \max\left(\frac{p}{100} \cdot L^n, \, 1\,\text{m}\right)\right] \tag{7}$$

Currently, we set $p = 10$.

### D.2    COLLISION RATE (CR)

The Collision Rate measures the fraction of agents that experience at least one collision with another agent during the prediction window. A collision is defined as any nonzero 3D Intersection-over-Union (IoU) between two predicted bounding boxes at the same timestep:

$$\text{CR} = \frac{1}{N} \sum_{n=1}^{N} \mathbf{1}\left[\exists t \in \{T_h + 1, \ldots, T\}, \, \exists m \neq n \text{ such that IoU}(B_n^{(t)}, B_m^{(t)}) > 0\right] \tag{8}$$

Here, $B_n^{(t)}$ and $B_m^{(t)}$ denote the predicted 3D bounding boxes for agents $n$ and $m$ at timestep $t$. The indicator function evaluates to 1 if agent $n$ overlaps with any other agent at any point in the future.

### D.3    OUT OF MAP RATE (OMR)

Let $\mathcal{M} \subset \mathbb{R}^2$ be the drivable area, defined via an $\alpha$-shape ($\alpha = 0.01$) over the top-down LiDAR point cloud. An agent is marked as out-of-map if any predicted $(x, y)$ position falls outside $\mathcal{M}$ during the forecast window:

$$\text{OMR} = \frac{1}{N} \sum_{n=1}^{N} \mathbf{1}\left[\exists t \in \{T_h + 1, \ldots, T\} \text{ such that } (\hat{t}_x^{(t)}, \hat{t}_y^{(t)}) \notin \mathcal{M}\right] \tag{9}$$

### D.4 FINAL DISPLACEMENT ERROR (FDE)

FDE measures the Euclidean distance between predicted and ground-truth centroids at the final timestep:

$$\text{FDE}^n = \left\| \hat{\mathbf{t}}^{(T)} - \mathbf{t}^{(T)} \right\|_2 \tag{10}$$

$$\text{FDE} = \frac{1}{N} \sum_{n=1}^{N} \text{FDE}^n \tag{11}$$

### D.5 AVERAGE DISPLACEMENT ERROR (ADE)

ADE averages the Euclidean error across all future timesteps:

$$\text{ADE}^n = \frac{1}{T_f} \sum_{t=T_h+1}^{T} \left\| \hat{\mathbf{t}}^{(t)} - \mathbf{t}^{(t)} \right\|_2 \tag{12}$$

$$\text{ADE} = \frac{1}{N} \sum_{n=1}^{N} \text{ADE}^n \tag{13}$$

### D.6 ROTATION ERROR (RE)

We define angular error per axis using cosine similarity, which handles angle periodicity (e.g., wrapping across $\pm\pi$):

$$\delta(\theta^{(t)}) = \cos^{-1}\left( \cos\theta^{(t)} \cdot \cos\hat{\theta}^{(t)} + \sin\theta^{(t)} \cdot \sin\hat{\theta}^{(t)} \right) \tag{14}$$

The agent-level rotation error is then:

$$\text{RE}^n = \frac{1}{3T_f} \sum_{t=T_h+1}^{T} \sum_{\theta \in \{\theta_{\text{yaw}}, \theta_{\text{pitch}}, \theta_{\text{roll}}\}} \delta(\theta^{(t)}) \tag{15}$$

$$\text{RE} = \frac{1}{N} \sum_{n=1}^{N} \text{RE}^n \tag{16}$$

### D.7 VELOCITY–HEADING SHIFT (VHS)

This metric measures misalignment between the direction of motion and the predicted yaw heading. Let $\hat{\mathbf{t}}^{(t)} = (\hat{t}_x^{(t)}, \hat{t}_y^{(t)}, \hat{t}_z^{(t)})$ be predicted centroids, and let $\hat{\boldsymbol{\theta}}^{(t)} = (\hat{\theta}_{\text{roll}}^{(t)}, \hat{\theta}_{\text{pitch}}^{(t)}, \hat{\theta}_{\text{yaw}}^{(t)})$ be predicted Euler angles. The following quantities are defined.

Planar velocity vector:
$$\mathbf{v}^{(t)} = \left( \hat{t}_x^{(t)} - \hat{t}_x^{(t-1)}, \ \hat{t}_y^{(t)} - \hat{t}_y^{(t-1)} \right) \tag{17}$$

Heading direction from yaw:
$$\mathbf{h}^{(t)} = \left( \cos\hat{\theta}_{\text{yaw}}^{(t)}, \ \sin\hat{\theta}_{\text{yaw}}^{(t)} \right) \tag{18}$$

Instantaneous angular deviation:

$$\delta^{(t)} = \cos^{-1}\left(\frac{\mathbf{v}^{(t)} \cdot \mathbf{h}^{(t)}}{\|\mathbf{v}^{(t)}\|_2}\right) \tag{19}$$

The per-agent VHS is the average of this angle over time:

$$\text{VHS}^n = \frac{1}{T_f - 1} \sum_{t=T_h+2}^{T} \delta^{(t)} \tag{20}$$

$$\text{VHS} = \frac{1}{N} \sum_{n=1}^{N} \text{VHS}^n \tag{21}$$

This formulation accounts for angle wrap-around using cosine-based angular comparison. Low VHS indicates physically plausible predictions (i.e., vehicles move in the direction they are instantaneously facing), making it a strong diagnostic of spatial reasoning quality.

## E  POLICY FOR MALFORMED OUTPUTS

Due to the generative nature of LLMs, predicted scene dictionaries occasionally contain malformed or inconsistent structures (e.g., misspelled keys, inhomogeneous arrays, mismatched dimensions). To ensure a fair and robust evaluation, we adopt a standardized repair policy before scoring predictions. Our procedure proceeds in three stages:

**(1) Key normalization.** We first correct common misspellings or synonymous key names to the canonical schema (`timestep`, `translation`, `rotation`, `size`, `attribute_label`). For example, `positions` → `translation`, `orientation` → `rotation`, and `time` → `timestep`.

**(2) Array validation and repair.** Predicted quantities (translations, rotations, etc.) are converted to NumPy arrays and checked for numerical type consistency. If an array contains non-numeric values or elements of inconsistent length:

- Inhomogeneous vectors are corrected via interpolation: the offending entry is replaced by the average of its nearest valid neighbors, or by the nearest valid element if only one neighbor exists.
- If no repair is possible, the entire instance is replaced by a *static fallback*, i.e., frozen at its last observed historical state.

We further enforce dimensional consistency: translations and rotations must be 3D, and timesteps must be 1D. Extra dimensions are removed, and singleton dimensions are expanded where necessary.

**(3) Temporal alignment.** All predicted attributes must have lengths consistent with the `timestep` field. If an attribute has a different length, it is resampled to match the temporal resolution via linear interpolation.

**(4) Final dictionary check.** After repairs, each scene dictionary is validated:

- All instance IDs must be strings, and all values must be dictionaries.
- Each instance must contain `timestep`, `translation`, and `rotation` fields, with arrays of correct type and shape (`translation`, `rotation` $\in \mathbb{R}^{T \times 3}$, `timestep` $\in \mathbb{Z}^T$).

If the dictionary fails this check, the entire predicted scene is replaced with a static fallback, constructed from the historical trajectories.

```
Input   : Predicted scene dictionary Ŝ, historical scene dictionary S
Output: Corrected scene dictionary S̃
foreach instance (i, Î_i) ∈ Ŝ do
    // Step 1:  Key normalization
    Replace all keys using predefined mapping
    // Step 2:  Array validation
    Convert values to arrays  if array invalid then
        Try interpolation  if repair fails then
            Î_i ← StaticFallback(S_i)  continue

    // Step 3:  Dimensionality checks
    Enforce correct shapes; remove extra dimensions  if invalid then
        Î_i ← StaticFallback(S_i)
    // Step 4:  Temporal alignment
    foreach attribute k do
        if len(Î_i[k]) ≠ len(Î_i[timestep]) then
            Resample Î_i[k]

// Step 5:  Final dictionary check
if validation fails then
    S̃ ← StaticFallback(S)
else
    S̃ ← Ŝ
return S̃
```

Algorithm 1: Malformed Output Policy. *StaticFallback*: A deterministic repair operation that replaces a malformed instance (or entire scene) with a static trajectory obtained by repeating the last available historical state across all future timesteps.

# F  EXAMPLES FROM REASONING OUTPUT

Listing 4 shows an example of a common tendency of reasoning models, i.e., assuming a constant velocity motion for non-static instances. This observation can explain the performance improvement of reasoning models with respect to their non-reasoning counterparts in the *linear* motion category, when cars' motion can indeed be well approximated by a constant-velocity model.

```
Alternatively, compute the average velocity over the last few
timesteps and project that. For instance, average Delta_x and
Delta_y over the last two timesteps (6-7 and 5-6):

Delta_x: (-3.499 + -3.474)/2 = -3.4865
Delta_y: (-3.738 + -3.76)/2 = -3.749

Using this average, forecast the next positions. But perhaps even
better to use the last observed velocity (timestep 7-8 would be
same as 6-7). However, looking at the trend, the Delta_x is
decreasing slightly each timestep (from -3.542 to -3.499 to
-3.474), which might indicate a deceleration. If this trend
continues, the Delta_x would decrease by about 0.025 each timestep
. Similarly, Delta_y is fluctuating but perhaps decreasing
slightly. However, without clear acceleration data, it's safer to
assume constant velocity for simplicity, ensuring smooth
trajectories.
```

Listing 4: Extract from the reasoning output of DeepSeek-R1 on scene `0aa4e8f5-2f9a-39a1-8f80-c2fdde4405a2_0_2`, explaining the assumption of constant velocity.

Listing 5 shows an example of the typical tendency of reasoning models to enumerate all instances in the history file and speculate about them, which can explain why they perform better than non-reasoning models in metrics such as Instance Precision and Recall.

```
Starting with the first instance, "1368947e-6555-47b2-95ba-
e24bfffd7b52". Looking at the translations, the x-coordinate
increases slightly each timestep [...]

Next, "b5edca23-fc00-4223-a1e8-f0572f70fca1" is a box truck. Its
translations show x decreasing slightly and y almost constant
[...]

Third instance "c836800b-2220-459c-8227-cca2d9b1a48b" has
translations where x decreases and y increases [...]

The fourth instance "1766d44d-c210-41e3-9047-048185004307" has
sparse timesteps. The translations show minor changes [...]

Fifth instance "3692678c-56b5-4f61-89b3-377ba7b96072" has
translations with x increasing slightly and y decreasing [...]

Last instance "e83eb243-bbdc-49a1-a3f2-c3f2999d1fa9" has
translations where x increases and y decreases rapidly [...]
```

Listing 5: Extract from the reasoning output of DeepSeek-R1 on scene `0b97f5dd-c396-3c02-b07d-b7fdbcb6c3d0_0_1`, showing the enumeration of existing instances.

## G  ADDITIONAL EXPERIMENTAL RESULTS

Tables 4, 5, 6, 7 show the full set of evaluation metrics for all experiments with pre-trained models across different motion categories (*overall*, *static*, *linear*, *nonlinear*, respectively). For distance-based metrics (i.e., ADD, ADE, and FDE), both the mean and the median over the test set are reported, while all remaining metrics are reported as the mean only.

Table 8 shows all evaluation metrics for the supervised fine-tuning experiment of Qwen3-4B in the *overall* category.

| Model | #Params | $\mathbf{ACC_f}\uparrow$ | ADD (mean)$\downarrow$ | ADD (median)$\downarrow$ | ADE (mean)$\downarrow$ | ADE (median)$\downarrow$ | FDE (mean)$\downarrow$ | FDE (median)$\downarrow$ |
|---|---|---|---|---|---|---|---|---|
| Linear baseline | | | **1.660** | 0.876 | **1.237** | 0.584 | **2.163** | 0.924 |
| Qwen3 (NR) | 14B | 0.995 | 1.792 | 0.594 | 1.459 | 0.454 | 2.469 | 0.830 |
| Qwen3 (R) | 14B | 0.977 | 2.237 | 0.666 | 1.803 | 0.503 | 2.966 | 0.865 |
| Llama-4 (NV) | 17B | 0.943 | 2.520 | 0.624 | 2.247 | 0.483 | 4.139 | 0.877 |
| Llama-4 (V) | 17B | 0.953 | 2.852 | 0.690 | 2.433 | 0.542 | 4.029 | 0.959 |
| gemma-3-it (NV) | 27B | **1.000** | 2.965 | 1.273 | 2.502 | 1.061 | 3.815 | 1.682 |
| gemma-3-it (V) | 27B | **1.000** | 3.291 | 1.464 | 2.797 | 1.228 | 4.340 | 1.918 |
| Qwen2.5-VL (NV) | 32B | 0.830 | 8.628 | 0.650 | 7.923 | 0.520 | 9.072 | 0.912 |
| Qwen2.5-VL (V) | 32B | 0.897 | 2.009 | 0.687 | 1.659 | 0.530 | 2.800 | 0.917 |
| Qwen3 (NR) | 32B | 0.942 | 1.965 | 0.595 | 1.612 | 0.447 | 2.668 | 0.816 |
| Qwen3 (R) | 32B | 0.918 | 16.228 | 0.653 | 13.893 | 0.488 | 15.147 | 0.860 |
| Qwen3 (NR) | 235B | 0.990 | 15.224 | **0.537** | 12.267 | **0.419** | 13.293 | **0.787** |
| Qwen3 (R) | 235B | 0.996 | 4.689 | 0.562 | 4.124 | 0.440 | 5.036 | 0.807 |
| DeepSeek-V3 | 671B | 0.999 | 1.882 | 0.598 | 1.536 | 0.471 | 2.647 | 0.842 |
| DeepSeek-R1 | 671B | 0.999 | 1.904 | 0.623 | 1.617 | 0.493 | 2.869 | 0.879 |
| Kimi-K2-Instruct | 1000B | 0.987 | 2.619 | 0.693 | 2.163 | 0.539 | 3.734 | 0.993 |

| Model | #Params | CR$\downarrow$ | $\mathbf{F1_{instance}}\uparrow$ | MR$\downarrow$ | OMR$\downarrow$ | $\mathbf{P_{instance}}\uparrow$ | RE$\downarrow$ | $\mathbf{R_{instance}}\uparrow$ | VHS$\downarrow$ |
|---|---|---|---|---|---|---|---|---|---|
| Linear baseline | | 0.050 | | 0.189 | 0.066 | | 0.030 | | 0.159 |
| Qwen3 (NR) | 14B | 0.036 | 0.916 | 0.152 | 0.068 | 0.862 | 0.016 | 0.977 | 0.108 |
| Qwen3 (R) | 14B | 0.039 | 0.924 | 0.166 | 0.069 | 0.871 | 0.027 | 0.985 | 0.098 |
| Llama-4 (NV) | 17B | 0.037 | 0.927 | 0.152 | 0.071 | 0.880 | 0.010 | 0.978 | 0.148 |
| Llama-4 (V) | 17B | 0.041 | 0.926 | 0.181 | 0.071 | 0.880 | 0.013 | 0.977 | 0.192 |
| gemma-3-it (NV) | 27B | 0.048 | 0.879 | 0.216 | **0.065** | 0.838 | 0.014 | 0.925 | 0.133 |
| gemma-3-it (V) | 27B | 0.048 | 0.878 | 0.227 | 0.068 | 0.838 | 0.015 | 0.921 | 0.158 |
| Qwen2.5-VL (NV) | 32B | 0.040 | 0.782 | 0.168 | 0.067 | 0.733 | 0.011 | 0.838 | 0.115 |
| Qwen2.5-VL (V) | 32B | 0.049 | 0.886 | 0.167 | 0.067 | 0.835 | 0.010 | 0.942 | 0.137 |
| Qwen3 (NR) | 32B | 0.039 | 0.927 | 0.155 | 0.071 | 0.873 | 0.014 | 0.988 | 0.149 |
| Qwen3 (R) | 32B | 0.038 | 0.925 | 0.162 | 0.070 | 0.874 | 0.015 | 0.981 | **0.090** |
| Qwen3 (NR) | 235B | 0.041 | 0.917 | 0.145 | 0.071 | 0.864 | 0.013 | 0.978 | 0.159 |
| Qwen3 (R) | 235B | 0.040 | 0.927 | **0.140** | 0.068 | 0.873 | 0.012 | 0.989 | 0.108 |
| DeepSeek-V3 | 671B | **0.034** | 0.935 | 0.152 | 0.070 | 0.879 | **0.009** | **0.999** | 0.116 |
| DeepSeek-R1 | 671B | 0.045 | **0.936** | 0.161 | 0.072 | **0.882** | 0.009 | 0.997 | 0.112 |
| Kimi-K2-Instruct | 1000B | 0.052 | 0.935 | 0.184 | 0.071 | 0.881 | 0.011 | 0.996 | 0.204 |

Table 4: Comparison of motion category **overall** across models. "R"/"NR" stands "reasoning"/"non-reasoning", "V"/"NV" stands for "vision"/"non-vision". Table split into two parts for layout purposes.

| Model | #Params | ADD (mean) ↓ | ADD (median) ↓ | ADE (mean) ↓ | ADE (median) ↓ |
|---|---|---|---|---|---|
| Linear baseline | | **0.487** | 0.295 | **0.281** | 0.193 |
| Qwen3 (NR) | 14B | 0.627 | 0.198 | 0.470 | **0.143** |
| Qwen3 (R) | 14B | 0.808 | 0.199 | 0.595 | 0.145 |
| Llama-4 (NV) | 17B | 1.795 | 0.226 | 1.683 | 0.164 |
| Llama-4 (V) | 17B | 1.688 | 0.238 | 1.460 | 0.181 |
| gemma-3-it (NV) | 27B | 1.072 | 0.263 | 0.881 | 0.195 |
| gemma-3-it (V) | 27B | 1.210 | 0.270 | 1.001 | 0.203 |
| Qwen2.5-VL (NV) | 32B | 5.377 | 0.211 | 4.732 | 0.159 |
| Qwen2.5-VL (V) | 32B | 0.798 | 0.226 | 0.646 | 0.171 |
| Qwen3-32B (NR) | 32B | 1.069 | 0.224 | 0.868 | 0.162 |
| Qwen3-32B (R) | 32B | 15.526 | 0.218 | 13.904 | 0.158 |
| Qwen3 (NR) | 235B | 16.639 | 0.214 | 13.243 | 0.159 |
| Qwen3 (R) | 235B | 4.236 | 0.217 | 3.633 | 0.163 |
| DeepSeek-V3 | 671B | 0.810 | **0.196** | 0.646 | 0.146 |
| DeepSeek-R1 | 671B | 1.135 | 0.217 | 0.985 | 0.167 |
| Kimi-K2-Instruct | 1000B | 2.054 | 0.259 | 1.705 | 0.193 |

| Model | #Params | FDE (mean) ↓ | FDE (median) ↓ | MR ↓ | OMR ↓ | RE ↓ | VHS ↓ |
|---|---|---|---|---|---|---|---|
| Linear baseline | | **0.445** | 0.289 | 0.109 | 0.061 | 0.019 | **0.047** |
| Qwen3 (NR) | 14B | 0.776 | 0.232 | 0.082 | 0.060 | 0.011 | 0.086 |
| Qwen3 (R) | 14B | 0.962 | **0.227** | 0.086 | 0.062 | 0.018 | 0.057 |
| Llama-4 (NV) | 17B | 3.106 | 0.302 | 0.093 | 0.064 | 0.004 | 0.159 |
| Llama-4 (V) | 17B | 2.343 | 0.309 | 0.115 | 0.063 | 0.005 | 0.186 |
| gemma-3-it (NV) | 27B | 1.170 | 0.287 | 0.104 | **0.060** | 0.009 | 0.078 |
| gemma-3-it (V) | 27B | 1.464 | 0.297 | 0.109 | 0.062 | 0.009 | 0.105 |
| Qwen2.5-VL (NV) | 32B | 4.970 | 0.256 | **0.074** | 0.061 | 0.005 | 0.094 |
| Qwen2.5-VL (V) | 32B | 1.036 | 0.281 | 0.093 | 0.060 | 0.004 | 0.118 |
| Qwen3-32B (NR) | 32B | 1.268 | 0.276 | 0.098 | 0.063 | 0.009 | 0.150 |
| Qwen3-32B (R) | 32B | 14.532 | 0.263 | 0.096 | 0.065 | 0.009 | 0.070 |
| Qwen3 (NR) | 235B | 13.675 | 0.270 | 0.097 | 0.065 | 0.008 | 0.143 |
| Qwen3 (R) | 235B | 3.961 | 0.277 | 0.089 | 0.062 | 0.006 | 0.108 |
| DeepSeek-V3 | 671B | 1.077 | 0.248 | 0.080 | 0.064 | **0.003** | 0.095 |
| DeepSeek-R1 | 671B | 1.684 | 0.284 | 0.106 | 0.067 | 0.003 | 0.096 |
| Kimi-K2-Instruct | 1000B | 2.844 | 0.340 | 0.139 | 0.065 | 0.005 | 0.220 |

Table 5: Comparison of motion category **static** across models. "R"/"NR" stands "reasoning"/"non-reasoning", "V"/"NV" stands for "vision"/"non-vision". Table split into two parts for layout purposes.

| Model | #Params | ADD (mean) ↓ | ADD (median) ↓ | ADE (mean) ↓ | ADE (median) ↓ |
|---|---|---|---|---|---|
| Linear baseline | | 4.409 | 3.168 | **3.364** | 2.299 |
| Qwen3 (NR) | 14B | 6.191 | 2.509 | 5.250 | 2.043 |
| Qwen3 (R) | 14B | 7.607 | 2.982 | 6.261 | 2.346 |
| Llama-4 (NV) | 17B | 5.291 | 2.112 | 4.436 | 1.604 |
| Llama-4 (V) | 17B | 5.749 | 2.199 | 4.780 | 1.688 |
| gemma-3-it (NV) | 27B | 11.106 | 7.171 | 9.547 | 6.215 |
| gemma-3-it (V) | 27B | 12.397 | 8.968 | 10.749 | 7.894 |
| Qwen2.5-VL (NV) | 32B | 32.158 | 3.034 | 31.436 | 2.443 |
| Qwen2.5-VL (V) | 32B | 5.897 | 2.591 | 4.950 | 2.194 |
| Qwen3-32B (NR) | 32B | 5.151 | 2.186 | 4.290 | 1.698 |
| Qwen3-32B (R) | 32B | 41.157 | 2.576 | 34.456 | 1.999 |
| Qwen3 (NR) | 235B | 14.985 | 1.950 | 12.744 | 1.498 |
| Qwen3 (R) | 235B | 5.686 | 1.926 | 5.069 | 1.504 |
| DeepSeek-V3 | 671B | 5.526 | 2.463 | 4.591 | 1.980 |
| DeepSeek-R1 | 671B | **4.108** | 2.075 | 3.392 | 1.632 |
| Kimi-K2-Instruct | 1000B | 4.163 | **1.895** | 3.431 | **1.468** |

| Model | #Params | FDE (mean) ↓ | FDE (median) ↓ | MR ↓ | OMR ↓ | RE ↓ | VHS ↓ |
|---|---|---|---|---|---|---|---|
| Linear baseline | | **5.644** | 4.106 | 0.302 | 0.081 | 0.066 | 0.187 |
| Qwen3 (NR) | 14B | 8.514 | 4.067 | 0.312 | 0.085 | 0.031 | 0.082 |
| Qwen3 (R) | 14B | 9.655 | 4.365 | 0.339 | 0.086 | 0.057 | 0.087 |
| Llama-4 (NV) | 17B | 7.227 | 3.236 | 0.247 | 0.090 | 0.023 | 0.098 |
| Llama-4 (V) | 17B | 7.791 | 3.490 | 0.266 | 0.088 | 0.036 | 0.115 |
| gemma-3-it (NV) | 27B | 14.782 | 9.692 | 0.605 | 0.079 | 0.021 | 0.119 |
| gemma-3-it (V) | 27B | 16.609 | 12.029 | 0.652 | **0.077** | 0.022 | 0.134 |
| Qwen2.5-VL (NV) | 32B | 35.403 | 4.767 | 0.395 | 0.091 | 0.028 | 0.094 |
| Qwen2.5-VL (V) | 32B | 8.236 | 4.198 | 0.308 | 0.089 | 0.020 | 0.105 |
| Qwen3-32B (NR) | 32B | 7.129 | 3.272 | 0.254 | 0.088 | 0.031 | 0.098 |
| Qwen3-32B (R) | 32B | 37.436 | 3.875 | 0.297 | 0.085 | 0.027 | 0.070 |
| Qwen3 (NR) | 235B | 15.413 | 3.115 | 0.219 | 0.088 | 0.026 | 0.111 |
| Qwen3 (R) | 235B | 7.420 | 3.050 | **0.199** | 0.081 | 0.024 | **0.052** |
| DeepSeek-V3 | 671B | 7.509 | 3.662 | 0.273 | 0.083 | **0.019** | 0.081 |
| DeepSeek-R1 | 671B | 5.673 | 3.185 | 0.210 | 0.080 | **0.019** | 0.080 |
| Kimi-K2-Instruct | 1000B | 5.758 | **2.993** | 0.221 | 0.083 | 0.024 | 0.099 |

Table 6: Comparison of motion category **linear** across models. "R"/"NR" stands "reasoning"/"non-reasoning", "V"/"NV" stands for "vision"/"non-vision". Table split into two parts for layout purposes.

| Model | #Params | ADD (mean) ↓ | ADD (median) ↓ | ADE (mean) ↓ | ADE (median) ↓ |
|---|---|---|---|---|---|
| Linear baseline | | 8.614 | 8.140 | 6.927 | 6.487 |
| Qwen3 (NR) | 14B | 5.510 | 4.465 | 4.545 | 3.564 |
| Qwen3 (R) | 14B | 7.550 | 6.058 | 6.147 | 4.909 |
| Llama-4 (NV) | 17B | 5.158 | 4.160 | 4.223 | 3.353 |
| Llama-4 (V) | 17B | 5.368 | 4.358 | 4.405 | 3.523 |
| gemma-3-it (NV) | 27B | 8.727 | 7.340 | 7.183 | 5.982 |
| gemma-3-it (V) | 27B | 9.467 | 7.890 | 7.854 | 6.285 |
| Qwen2.5-VL (NV) | 32B | 16.089 | 4.760 | 13.673 | 3.965 |
| Qwen2.5-VL (V) | 32B | 5.477 | 4.600 | 4.425 | 3.662 |
| Qwen3-32B (NR) | 32B | 4.605 | 3.795 | 3.658 | 2.989 |
| Qwen3-32B (R) | 32B | 14.039 | 4.706 | 12.526 | 3.859 |
| Qwen3 (NR) | 235B | 23.053 | **3.341** | 17.648 | **2.543** |
| Qwen3 (R) | 235B | 4.374 | 3.830 | 3.469 | 3.056 |
| DeepSeek-V3 | 671B | 5.296 | 4.449 | 4.260 | 3.672 |
| DeepSeek-R1 | 671B | 4.836 | 4.198 | 3.911 | 3.438 |
| Kimi-K2-Instruct | 1000B | **4.149** | 3.413 | **3.290** | 2.843 |

| Model | #Params | FDE (mean) ↓ | FDE (median) ↓ | MR ↓ | OMR ↓ | RE ↓ | VHS ↓ |
|---|---|---|---|---|---|---|---|
| Linear baseline | | 13.343 | 12.618 | 0.996 | 0.039 | 0.102 | 0.348 |
| Qwen3 (NR) | 14B | 9.178 | 7.906 | 0.761 | 0.047 | 0.065 | 0.164 |
| Qwen3 (R) | 14B | 12.059 | 10.148 | 0.890 | 0.048 | 0.103 | 0.208 |
| Llama-4 (NV) | 17B | 8.785 | 7.368 | 0.746 | 0.050 | 0.062 | 0.166 |
| Llama-4 (V) | 17B | 8.653 | 7.909 | 0.765 | 0.043 | 0.066 | 0.180 |
| gemma-3-it (NV) | 27B | 12.409 | 10.878 | 0.890 | **0.038** | 0.075 | 0.168 |
| gemma-3-it (V) | 27B | 13.487 | 11.894 | 0.901 | **0.038** | 0.079 | 0.175 |
| Qwen2.5-VL (NV) | 32B | 18.487 | 8.343 | 0.780 | 0.041 | 0.063 | 0.172 |
| Qwen2.5-VL (V) | 32B | 9.074 | 8.061 | 0.773 | 0.041 | 0.060 | 0.184 |
| Qwen3-32B (NR) | 32B | 7.831 | 6.773 | 0.678 | 0.050 | 0.062 | 0.207 |
| Qwen3-32B (R) | 32B | 17.361 | 8.062 | 0.782 | 0.043 | 0.067 | 0.161 |
| Qwen3 (NR) | 235B | 21.481 | **5.890** | **0.614** | 0.052 | 0.059 | 0.213 |
| Qwen3 (R) | 235B | 7.721 | 7.040 | 0.712 | 0.045 | 0.058 | 0.188 |
| DeepSeek-V3 | 671B | 8.777 | 8.000 | 0.781 | 0.042 | 0.056 | **0.157** |
| DeepSeek-R1 | 671B | 8.396 | 7.690 | 0.751 | 0.045 | **0.053** | 0.177 |
| Kimi-K2-Instruct | 1000B | **7.226** | 6.260 | 0.627 | 0.048 | 0.054 | 0.178 |

Table 7: Comparison of motion category **nonlinear** across models. "R"/"NR" stands "reasoning"/"non-reasoning", "V"/"NV" stands for "vision"/"non-vision". Table split into two parts for layout purposes.

| Model | #Params | $ACC_f$ ↑ | ADD (mean) ↓ | ADD (median) ↓ | ADE (mean) ↓ | ADE (median) ↓ | FDE (mean) ↓ | FDE (median) ↓ |
|---|---|---|---|---|---|---|---|---|
| Qwen3-4B pre-trained | 4B | 0.943 | **4.193** | **0.688** | **3.609** | **0.516** | **4.867** | **0.906** |
| Qwen3-4B fine-tuned | 4B | **0.995** | 5.523 | 0.956 | 4.721 | 0.775 | 6.089 | 1.254 |

| Model | #Params | CR ↓ | $F1_{instance}$ ↑ | MR ↓ | OMR ↓ | $P_{instance}$ ↑ | RE ↓ | $R_{instance}$ ↑ | VHS ↓ |
|---|---|---|---|---|---|---|---|---|---|
| Qwen3-4B pre-trained | 4B | 0.042 | 0.914 | **0.167** | 0.071 | 0.862 | 0.012 | 0.974 | **0.091** |
| Qwen3-4B fine-tuned | 4B | **0.034** | **0.928** | 0.188 | **0.066** | **0.875** | **0.011** | **0.989** | 0.138 |

Table 8: Comparison of motion category **overall** between pre-trained and fine-tuned Qwen3-4B. Table split into two parts for layout purposes.

