# OpenReview forum: "Car4Cast: A Dataset and Benchmark for LLM-Based Motion Forecasting and Spatial Reasoning in Autonomous Driving"
_ICLR.cc/2026/Conference — Submitted to ICLR 2026_

### Official Review · Reviewer_H1Lv · 2025-10-25

**Soundness:** 3
**Presentation:** 4
**Contribution:** 3
**Rating:** 6
**Confidence:** 4

**Summary:**

This paper introduces a new dataset designed to benchmark large language models (LLMs) as the prediction and planning module in autonomous driving. The authors curate scenarios from established datasets, add annotations, and provide an additional bird’s-eye-view (BEV) LiDAR input. They also benchmark several LLMs alongside a simple linear baseline, analyzing results across object categories and motion types (linear vs. nonlinear).

The paper is well written and provides timely contributions to the ongoing discussion about whether, and how, LLMs should be integrated into autonomous driving pipelines. However, the current framing feels somewhat too specialized toward spatial reasoning tasks. Making the dataset more broadly applicable would strengthen its utility.

**Strengths:**

The dataset is a valuable contribution, addressing the gap of standardized benchmarks for LLMs in autonomous driving.

The benchmarking is thorough, and the inclusion of a linear baseline is well chosen and informative.

**Weaknesses:**

While the paper is strong overall, I see three notable weaknesses:

No map input is included, even though maps are a standard input in many non-LLM models.

Camera inputs are missing. Their inclusion would enable cross-comparison with end-to-end (E2E) models and make the dataset more versatile.

The reproducibility of evaluation is unclear. Without a standardized evaluation suite, adoption by the community may be limited.

**Questions:**

Addressing at least two of the following questions would raise my score, with more resolved leading to a higher increase:

Can maps be provided as an additional input?

Can camera data be included as well?

How will evaluation be standardized? For example, will a machine-readable format (e.g., JSON, as in nuScenes) and evaluation toolkit be provided? Otherwise, adoption might face significant barriers.

The linear baseline is a great choice. Could you also include a simple 2-layer MLP baseline to further strengthen the results?

---

> ### Author Response · Authors · 2025-11-21
>
> Thank you for the constructive feedback and for the understanding of our goals and rationale in developing Car4Cast. Below we address the posed questions, while remaining fully available for further clarifications.
>
> 1. Can maps be provided as an additional input?
>
> When available in the source datasets, **including maps is fully feasible**. For demonstration, we collected and rendered **semantic road maps** from the “mini” split of nuScenes and added them as an **image prompt** to the VLMs tested in our paper. An example of such map can be found in the nuScenes documentation (https://www.nuscenes.org/nuscenes?tutorial=maps). **We will add all available maps in our final version.**
>
> We evaluated VLM performance on nuScenes-mini in three settings: **no map input, LiDAR top-down map (as in the paper), and semantic map input**.
>
> | Model                             | # Params | ACC_f ↑  | ADD (mean) ↓ | ADD (median) ↓ | F1_instance ↑ | P_instance ↑ | R_instance ↑ |
> |-----------------------------------|----------|----------|--------------|----------------|---------------|--------------|--------------|
> | Llama-4-Maverick-17B              | 17B      | 0.947    | 2.690        | 0.877          | 0.932         | **0.889**    | 0.979        |
> | Llama-4-Maverick-17B (LiDAR map) | 17B      | 0.976    | **2.593**    | **0.792**      | **0.938**     | 0.886        | **0.997**    |
> | Llama-4-Maverick-17B (semantic map) | 17B    | **0.984** | 59.207      | 0.973          | 0.936         | **0.889**    | 0.987        |
> | | | | | | | | |
> | gemma-3-27b-it                    | 27B      | 0.997    | 5.907        | 2.980          | 0.803         | 0.802        | 0.803        |
> | gemma-3-27b-it (LiDAR map)        | 27B      | **1.000** | 5.982       | **2.671**      | 0.831         | 0.829        | 0.833        |
> | gemma-3-27b-it (semantic map)     | 27B      | **1.000** | **5.665**   | 2.721          | **0.846**     | **0.842**    | **0.851**    |
> | | | | | | | | |
> | Qwen2.5-VL-32B                    | 32B      | 0.996    | 75.004       | 1.652          | 0.934         | 0.884        | 0.990        |
> | Qwen2.5-VL-32B (LiDAR map)        | 32B      | **1.000** | **2.756**   | **0.830**      | 0.935         | 0.887        | 0.988        |
> | Qwen2.5-VL-32B (semantic map)     | 32B      | **1.000** | 9.448       | 1.154          | **0.938**     | **0.888**    | **0.993**    |
>
> The results with the semantic map are consistent with the findings reported in our paper: they **improve formatting metrics** such as Formatting Accuracy and Instance Precision/Recall, while **geometric metrics show different trends**. This observation highlights an open research question about VLM performance in structured spatial reasoning. As a dataset and benchmark paper, a key goal is to **reveal such outcomes to guide future research**.
>
> 2. Can camera data be included as well?
>
> Similarly, we **collected camera images from the source datasets** and paired them with corresponding trajectories and timestamps. For simplicity, we used only the **front-facing camera**, but the same process can be extended to multiple views supported by source datasets. **We will add the minimum common set of available camera views from all source datasets in our final version.**
>
> | Model                           | # Params | ACC_f ↑  | ADD (mean) ↓ | ADD (median) ↓ | F1_instance ↑ | P_instance ↑ | R_instance ↑ |
> |---------------------------------|----------|----------|--------------|----------------|---------------|--------------|--------------|
> | Llama-4-Maverick-17B            | 17B      | 0.947    | 2.690        | 0.877          | 0.932         | 0.889        | 0.979        |
> | Llama-4-Maverick-17B (with camera) | 17B   | **0.994** | **2.436**   | **0.769**      | **0.939**     | **0.895**    | **0.987**    |
> | | | | | | | | |
> | gemma-3-27b-it                  | 27B      | 0.997    | 5.907        | 2.980          | 0.803         | 0.802        | **0.803**    |
> | gemma-3-27b-it (with camera)    | 27B      | **1.000** | **5.788**   | **2.429**      | **0.817**     | **0.845**    | 0.791        |
> | | | | | | | | |
> | Qwen2.5-VL-32B                  | 32B      | **0.996** | 75.004      | 1.652          | **0.934**     | 0.884        | **0.990**    |
> | Qwen2.5-VL-32B (with camera)    | 32B      | 0.936    | **4.131**    | **1.596**      | 0.931         | **0.892**    | 0.972        |
>
> Evaluation on nuScenes-mini shows that including camera input **improves geometric metrics** (e.g., ADD), indicating that VLMs capture meaningful information from the camera modality, and highlighting the potential value of multi-view extensions in future research.
>
> 3. How will evaluation be standardized?
>
> Our evaluation suite is **included in the supplementary code**. **Any model** can be evaluated by outputting predictions in the **same structured JSON format** as the inputs. The evaluation toolkit then computes all metrics automatically, ensuring reproducibility and facilitating community adoption.

---

> > ### Author Response · Authors · 2025-11-21
> >
> > 4. Could you also include a simple 2-layer MLP baseline?
> >
> > We implemented a **two-layer MLP** (with layer sizes 48, 1024, 48) predicting trajectory residuals relative to a first-last point linear extrapolation of historical positions. Additionally, we implemented a 12D constant-velocity **Kalman Filter** with a state vector including 3D position, velocity, rotation, and angular velocity, with diagonal process and measurement noise covariances.
> >
> > Below we show a comparison of motion category *overall* across models on the test set.
> >
> > | Model                         | # Params | ACC_f ↑ | ADD (mean) ↓ | ADD (median) ↓ | CR ↓  | F1_instance | MR ↓   | OMR ↓   | P_instance ↑ | RE ↓   | R_instance ↑ |
> > |-------------------------------|----------|---------|--------------|----------------|-------|-------------|--------|---------|--------------|--------|--------------|
> > | Linear baseline               |          |         | 1.660        | 0.876          | 0.050 |    | 0.189  | 0.066   |              | 0.030  |
> > | 2-layer MLP                   | 99376    |         | 1.528        | 0.782          | 0.048 |   | 0.169  | 0.066   |              | 0.030  |
> > | Kalman Filter                 |          |         | **1.194**    | **0.527**      | 0.036 |  | **0.130** | 0.066 |              | 0.010  |
> > | Qwen3-14B (non-reasoning)    | 14B      | 0.995   | 1.792        | 0.594          | 0.036 | 0.916       | 0.152  | 0.068   | 0.862        | 0.016  | 0.977        |
> > | Qwen3-14B (reasoning)        | 14B      | 0.977   | 2.237        | 0.666          | 0.039 | 0.924       | 0.166  | 0.069   | 0.871        | 0.027  | 0.985        |
> > | Llama-4-Maverick-17B (non-vision) | 17B | 0.943   | 2.520        | 0.624          | 0.037 | 0.927       | 0.152  | 0.071   | 0.880        | 0.010  | 0.978        |
> > | Llama-4-Maverick-17B (vision)| 17B      | 0.953   | 2.852        | 0.690          | 0.041 | 0.926       | 0.181  | 0.071   | 0.880        | 0.013  | 0.977        |
> > | gemma-3-27b-it (non-vision)  | 27B      | **1.000**   | 2.965        | 1.273          | 0.048 | 0.879       | 0.216  | **0.065** | 0.838      | 0.014  | 0.925        |
> > | gemma-3-27b-it (vision)      | 27B      | **1.000**   | 3.291        | 1.464          | 0.048 | 0.878       | 0.227  | 0.068   | 0.838        | 0.015  | 0.921        |
> > | Qwen2.5-VL-32B (non-vision)  | 32B      | 0.830   | 8.628        | 0.650          | 0.040 | 0.782       | 0.168  | 0.067   | 0.733        | 0.011  | 0.838        |
> > | Qwen2.5-VL-32B (vision)      | 32B      | 0.897   | 2.009        | 0.687          | 0.049 | 0.886       | 0.167  | 0.067   | 0.835        | 0.010  | 0.942        |
> > | Qwen3-32B (non-reasoning)    | 32B      | 0.942   | 1.965        | 0.595          | 0.039 | 0.927       | 0.155  | 0.071   | 0.873        | 0.014  | 0.988        |
> > | Qwen3-32B (reasoning)        | 32B      | 0.918   | 16.228       | 0.653          | 0.038 | 0.925       | 0.162  | 0.070   | 0.874        | 0.015  | 0.981        |
> > | Qwen3-235B-A22B (non-reasoning)| 235B    | 0.990   | 15.224       | 0.537          | 0.041 | 0.917       | 0.145  | 0.071   | 0.864        | 0.013  | 0.978        |
> > | Qwen3-235B-A22B (reasoning)  | 235B     | 0.996   | 4.689        | 0.562          | 0.040 | 0.927       | 0.140  | 0.068   | 0.873        | 0.012  | 0.989        |
> > | DeepSeek-V3                   | 671B     | 0.999   | 1.882        | 0.598          | 0.034 | 0.935       | 0.152  | 0.070   | 0.879        | **0.009** | 0.999      |
> > | DeepSeek-R1                   | 671B     | 0.999   | 1.904        | 0.623          | 0.045 | **0.936**       | 0.161  | 0.072   | **0.882**    | 0.009  | 0.997        |
> > | Kimi-K2-Instruct              | 1000B    | 0.987   | 2.619        | 0.693          | 0.052 | 0.935       | 0.184  | 0.071   | 0.881        | 0.011  | 0.996        |
> > | Qwen3-4B pre-trained          | 4B       | 0.943   | 4.193        | 0.688          | 0.042 | 0.914       | 0.167  | 0.071   | 0.862        | 0.012  | 0.974        |
> > | Qwen3-4B fine-tuned           | 4B       | 0.995   | 5.523        | 0.956          | **0.034** | 0.928   | 0.188  | 0.066   | 0.875        | 0.011  | 0.989        |
> >
> > **Both baselines outperform the linear baseline (hence all tested models)** when averaged over the full test set, providing **stronger reference points** and **reinforcing our findings** on the challenges current LLMs and VLMs face in structured spatial prediction.
> >
> >
> > We agree that adopting your suggestions can broaden the scope and applicability of our dataset. Our evaluations show that both map and camera modalities can be integrated effectively, and we hope our responses have addressed all questions in a clear and satisfactory manner.

---

> > > ### Comment · Reviewer_H1Lv · 2025-11-25
> > > **Rebuttal**
> > >
> > > Thank you for the valuable edits . I think this paper is extremely valuable to the community. Pls make sure the edits are also presented in the camera ready version of the paper. I raised to 10

---

### Official Review · Reviewer_Fw34 · 2025-10-28

**Soundness:** 2
**Presentation:** 1
**Contribution:** 2
**Rating:** 2
**Confidence:** 3

**Summary:**

This paper introduces Car4Cast, which casts 3D motion forecasting in autonomous driving as a spatial reasoning task. And it provides a unified evaluation suite with 3D forecasting and LLM-specific structured prediction metrics.

**Strengths:**

It provides 1,000 scenes and over 100,000 trajectories from established 3D autonomous driving datasets. It provides a visual modality in the form of top-down LiDAR-based maps, enabling evaluation with Vision-Language Models (VLMs) and supporting multimodal extensions.

**Weaknesses:**

1. The paper introduces 3D motion forecasting in autonomous driving as a spatial reasoning task; however, the incorporation of Large Language Models (LLMs) introduces hallucination issues. The paper does not clearly explain the rationale for framing 3D motion forecasting as a spatial reasoning task, nor does it elaborate on the advantages of this formulation compared to traditional approaches. As a result, it is difficult to assess the novelty and contribution of the work.
2. The paper lacks comparisons with conventional motion forecasting methods, so how to prove that LLM/VLM is effective?
3. The paper lacks visual analyses of motion forecasting results across different models. As a benchmark, it needs more details about the evaluation.

**Questions:**

Same to weaknesses.

---

> ### Author Response · Authors · 2025-11-21
>
> Thank you for the insightful feedback. We address each of the concerns below.
>
> 1. Rationale for framing 3D motion forecasting as a spatial reasoning task
>
> We appreciate the opportunity to clarify this. We will expand the explanation in the introduction accordingly.
>
> Motion forecasting is particularly **well-suited for evaluating spatial reasoning** because it
> - requires grounding **numerical inputs** in **physical quantities in space**,
> - demands understanding of trajectories in the **spatial and temporal** dimensions,
> - requires accounting for **interactions between multiple-agents**,
> - provides precise **quantitative supervision**, allowing for a standardized numerical evaluation,
> - is **intuitive for humans but non-trivial for LLMs**, making it a revealing probe into their inductive biases and reasoning strategies.
>
> These characteristics make forecasting a suitable task to **isolate where LLMs fail spatially**, how they **reason geometrically**, and which aspects of the task challenge current training paradigms.
>
> While not the main goal of the paper, the use of language models for motion forecasting also brings several secondary benefits compared to traditional architectures:
> - **Explainability** : Predictions are accompanied by **natural-language** explanations, helping humans inspect intermediate reasoning patterns or errors.
> - **Interactivity**: LLMs enable programmable or **conversational interfaces** that can incorporate constraints, scenario descriptions, or high-level instructions.
> - **Knowledge-rich priors**: Large pre-trained models encode extensive **conceptual knowledge about traffic norms**, object interactions, and general physical logics, which are difficult to inject into conventional models.

---

> > ### Author Response · Authors · 2025-11-21
> >
> > 2. Comparisons with conventional motion forecasting methods
> >
> > We agree that **comparing against classical baselines strengthens the benchmark**. In addition to the linear model in the main paper, we implemented **two further baselines**:
> > - A **two-layer MLP** (with layer sizes 48, 1024, 48) that predicts trajectory residuals relative to a first–last point linear extrapolation of historical motion.
> > - A 12-D constant-velocity **Kalman Filter**, whose state includes 3D position, velocity, rotation, and angular velocity; the measurement model observes 3D position and rotation, with diagonal process/measurement covariances and rotation handled via wrapped angles.
> >
> > Below we show a comparison of motion category *overall* across models on the test set.
> >
> > | Model                         | # Params | ACC_f ↑ | ADD (mean) ↓ | ADD (median) ↓ | CR ↓  | F1_instance | MR ↓   | OMR ↓   | P_instance ↑ | RE ↓   | R_instance ↑ |
> > |-------------------------------|----------|---------|--------------|----------------|-------|-------------|--------|---------|--------------|--------|--------------|
> > | Linear baseline               |          |         | 1.660        | 0.876          | 0.050 |    | 0.189  | 0.066   |              | 0.030  |
> > | 2-layer MLP                   | 99376    |         | 1.528        | 0.782          | 0.048 |   | 0.169  | 0.066   |              | 0.030  |
> > | Kalman Filter                 |          |         | **1.194**    | **0.527**      | 0.036 |  | **0.130** | 0.066 |              | 0.010  |
> > | Qwen3-14B (non-reasoning)    | 14B      | 0.995   | 1.792        | 0.594          | 0.036 | 0.916       | 0.152  | 0.068   | 0.862        | 0.016  | 0.977        |
> > | Qwen3-14B (reasoning)        | 14B      | 0.977   | 2.237        | 0.666          | 0.039 | 0.924       | 0.166  | 0.069   | 0.871        | 0.027  | 0.985        |
> > | Llama-4-Maverick-17B (non-vision) | 17B | 0.943   | 2.520        | 0.624          | 0.037 | 0.927       | 0.152  | 0.071   | 0.880        | 0.010  | 0.978        |
> > | Llama-4-Maverick-17B (vision)| 17B      | 0.953   | 2.852        | 0.690          | 0.041 | 0.926       | 0.181  | 0.071   | 0.880        | 0.013  | 0.977        |
> > | gemma-3-27b-it (non-vision)  | 27B      | **1.000**   | 2.965        | 1.273          | 0.048 | 0.879       | 0.216  | **0.065** | 0.838      | 0.014  | 0.925        |
> > | gemma-3-27b-it (vision)      | 27B      | **1.000**   | 3.291        | 1.464          | 0.048 | 0.878       | 0.227  | 0.068   | 0.838        | 0.015  | 0.921        |
> > | Qwen2.5-VL-32B (non-vision)  | 32B      | 0.830   | 8.628        | 0.650          | 0.040 | 0.782       | 0.168  | 0.067   | 0.733        | 0.011  | 0.838        |
> > | Qwen2.5-VL-32B (vision)      | 32B      | 0.897   | 2.009        | 0.687          | 0.049 | 0.886       | 0.167  | 0.067   | 0.835        | 0.010  | 0.942        |
> > | Qwen3-32B (non-reasoning)    | 32B      | 0.942   | 1.965        | 0.595          | 0.039 | 0.927       | 0.155  | 0.071   | 0.873        | 0.014  | 0.988        |
> > | Qwen3-32B (reasoning)        | 32B      | 0.918   | 16.228       | 0.653          | 0.038 | 0.925       | 0.162  | 0.070   | 0.874        | 0.015  | 0.981        |
> > | Qwen3-235B-A22B (non-reasoning)| 235B    | 0.990   | 15.224       | 0.537          | 0.041 | 0.917       | 0.145  | 0.071   | 0.864        | 0.013  | 0.978        |
> > | Qwen3-235B-A22B (reasoning)  | 235B     | 0.996   | 4.689        | 0.562          | 0.040 | 0.927       | 0.140  | 0.068   | 0.873        | 0.012  | 0.989        |
> > | DeepSeek-V3                   | 671B     | 0.999   | 1.882        | 0.598          | 0.034 | 0.935       | 0.152  | 0.070   | 0.879        | **0.009** | 0.999      |
> > | DeepSeek-R1                   | 671B     | 0.999   | 1.904        | 0.623          | 0.045 | **0.936**       | 0.161  | 0.072   | **0.882**    | 0.009  | 0.997        |
> > | Kimi-K2-Instruct              | 1000B    | 0.987   | 2.619        | 0.693          | 0.052 | 0.935       | 0.184  | 0.071   | 0.881        | 0.011  | 0.996        |
> > | Qwen3-4B pre-trained          | 4B       | 0.943   | 4.193        | 0.688          | 0.042 | 0.914       | 0.167  | 0.071   | 0.862        | 0.012  | 0.974        |
> > | Qwen3-4B fine-tuned           | 4B       | 0.995   | 5.523        | 0.956          | **0.034** | 0.928   | 0.188  | 0.066   | 0.875        | 0.011  | 0.989        |
> >
> > **Both baselines outperform the linear model (hence all tested LLMs)** when averaged across the full test set, highlighting that they are substantially stronger and offering harder points of comparison for future approaches.
> >
> > At the same time, these results reinforce the **main takeaway of our experimental findings: LLMs/VLMs currently perform worse than even simple forecasting methods, so naturally they also fall short of more tailored motion-forecasting methods. This gap is precisely the capability deficiency that Car4Cast is designed to quantify and narrow.**

---

> > > ### Author Response · Authors · 2025-11-28
> > >
> > > 3. Visual analysis
> > >
> > > We agree that visual analysis is highly valuable for understanding model behavior under our benchmark. In response, **we provide qualitative video visualizations** showing historical trajectories, predicted future trajectories, and ground-truth futures, to **support the observations we raised in our Experiments section** and show cases where they actually occur. These visualizations are now included **in the `visual_analysis` folder of our supplementary material** (organized by `scene_id`).
> > >
> > > Below we summarize the aforementioned observations and link them to visual examples:
> > >
> > > **• LLMs underperform a linear baseline in the mean case due to catastrophic failures**
> > >
> > > We observe that mean ADD is strongly driven by **catastrophic failures on some scenes**. For example, Qwen3-235B-A22B in non-reasoning mode produces an **ADD exceeding 4000** in scene `ea6895f2-504b-37b5-bfd0-cbf7017f22c3_1_1`, where all instances are incorrectly predicted as stationary at a location notably **far from both historical and ground-truth trajectories**. These qualitatively visible failures explain why models show a discrepancy between median and mean metrics.
> > >
> > > **• Reasoning models outperform non-reasoning models in the *linear* motion category** (for ≥235B parameters)
> > >
> > > We selected scenes containing only linearly moving vehicles and compared predictions by corresponding reasoning and non-reasoning models.
> > >
> > > For example, in scene `01452fbfbf4543af8acdfd3e8a1ee806_1_0`, a bus appears for only one historical timestep:
> > >
> > > **DeepSeek-V3** extrapolates its motion in the **wrong direction**,  while **DeepSeek-R1** correctly infers motion aligned with the vehicle’s heading.
> > >
> > > A similar pattern appears for **Qwen3-235B-A22B**, where the **non-reasoning variant** chooses the **wrong motion direction**, whereas the reasoning variant opts for a static forecast, at least consistent with the heading.
> > >
> > > A further example is scene `0b1b993a-68b3-3232-9afa-fc9942b5b79b_0_4`, where the reasoning version of Qwen3-235B-A22B produces a realistic linear extrapolation for a vehicle observed for only two timesteps, whereas the **non-reasoning model overestimates its velocity**. These observations suggest that, in general, **reasoning models are more capable of extrapolating a linear motion from few historical observations**, which turns out to be correct when the motion of the instance is in fact linear.
> > >
> > > **• VLMs show better Formatting Accuracy when image input is provided**
> > >
> > > We take **Llama-4-Maverick-17B-128E-Instruct-FP8** as an example.
> > > In scenes `fd4ef697de684d0a8e016a8f8ae61193_0_1` and `f9f6a7e9-4f79-3fdf-b1a7-ba300622f116_0_2`,  the highlighted moving cars are **incorrectly formatted by the non-vision model**, so they are **assumed to be static** at their last known historical position for evaluation and visualization purposes. The variant with visual input instead produces cleanly formatted predictions.
> > >
> > > Even with these wrong-formatting penalties, the **text-only model achieves lower median ADD** across all motion categories (*static*, *linear*, and *nonlinear*), indicating a substantial performance degrade when introducing the LiDAR map modality.
> > >
> > > **• Supervised Fine-Tuning improves formatting but harms geometric accuracy**
> > >
> > > Visualizations also clarify the misalignment introduced by SFT in our experiment with Qwen3-4B.
> > >
> > > In scene `d3d94f2ce3dc4db4b3ba6f4aa81c3987_1_4`, the pre-trained model misformats some parked cars, which are safely treated as static. The **SFT model produces perfectly formatted JSON** (ACC_f = 1) but predicts **trajectories in a completely incorrect region**, causing a significant increase in ADD.
> > >
> > > In scene `285ac213-8caf-31a4-b0fa-c240580f7f69_0_3`, the **SFT model** shows again a higher Formatting Accuracy but **forecasts a vehicle moving in the opposite direction** from its historical trajectory (unlike the previously presented case, the historical trajectory of the car is even gap-free and quite stable).
> > >
> > > Once again, even with the static fallback given to unreadable instance, **the non-finetuned model performs better in terms of median ADD across all motion categories (*static*, *linear*, *nonlinear*)**, indicating a substantial performance degrade due to supervised fine-tuning and highlighting the need for more suitable training paradigms for quantitative spatial intelligence.
> > >
> > > ---
> > >
> > > **We will add this analysis to the final version of our paper.**
> > >
> > > Thank you once again for your feedback, we hope our explanations and proposed integrations will help strengthen our benchmark and clarify our objectives. Don’t hesitate to ask if further insights are needed.

---

### Official Review · Reviewer_fbaA · 2025-10-31

**Soundness:** 2
**Presentation:** 3
**Contribution:** 2
**Rating:** 4
**Confidence:** 4

**Summary:**

This paper introduces Car4Cast, a dataset and benchmark that reframes 3D motion forecasting as a structured text generation problem for LLMs/VLMs. The benchmark includes over 11k scenarios from a collection of AD datasets (nuScenes, KITTI, PandaSet, Argoverse 2, CADC). It provides both textual inputs and an optional visual modality, and designed LLM-oriented evaluation metrics like formatting accuracy and hallucinated objects percentage. Experiments across LLMs/VLMs plus SFT on Qwen3-4B show: (1) LLMs generally perform better than a linear baseline but show worse average result due to some significant errors in some cases; (2) VLMs perform even worse when extra maps are provided; (3) SFT improves answer format but decreases numerical accuracy, indicating misalignment between token-level objectives and numerical correctness.

**Strengths:**

1.	The writing is clear. The proposed benchmark and contributions are easy to understand. Formatting multi-agent 3D forecasting as structured language output is novel, and the correspond evaluation metric design beyond the traditional ADE/FDE is reasonable.
2.	The experiments cover a wide range of LLMs/VLMs with different model size/families, and thus eliminate the bias caused by individual models and shows informative results. A negative result on SFT is also informative and honest.
3.	Processing pipeline, dataset, evaluation code, prompts and training setups are all provided, so this dataset is ready to be used and the pipeline is easy to reproduce.

**Weaknesses:**

1.	Regarding the contribution of this benchmark, most of the evaluations proposed in the paper can actually be accomplished on the nuScenes or KITTI datasets themselves through interface definitions + prompt design, not requiring much additional manual effort. The primary contributions regarding the dataset are merely the combination of datasets and the extra metrics.
2.	The SFT result highlights cross-entropy’s misalignment with numerical/geometry accuracy, but other benchmarks like NuScenes-SpatialQA has shown some LLM’s ability to perform spatial inference. Why does it get even worse but not at least remain at a similar level? No solution that could possibly improve the performance is provided (e.g., structured losses by separating non-numerical tokens and numbers). If the dataset cannot improve the performance of motion forecasting, the persuasiveness of the contribution will be further diminished.
3.	The reason of non-reasoning models overperforming reasoning models in general remains unclear. Given that CoT has demonstrated strong performance in other logical reasoning problems, it is puzzling.

**Questions:**

1.	Is the benchmark also compatible for LLM-based models? Since the pure LLMs/VLMs perform poorly in the benchmark, is it possible to include recent language-based forecasting systems (e.g., TrajLLM) to show the validity and practical significance of the benchmark?
2.	The experiments for VLMs: The image prompt is just using a simple point-cloud-transformed BEV image, and point clouds sometimes contain noises (By checking the provided map images, noises are indeed observed). Is it possible to gain better results by using RGB BEV images? Or by denoising and increase the resolution of the image?

---

> ### Author Response · Authors · 2025-11-21
>
> Thank you for the detailed feedback and insightful questions. Below we address the concerns raised.
>
> 1) Contribution of the dataset beyond combining existing datasets:
>
> While Car4Cast aggregates multiple established datasets, the **data extraction and processing required substantial effort**. High-level pseudocode of our pipeline:
> ```
> for each source dataset:
>     parse the dataset’s annotation format (varies across datasets)
>     for each scene:
>         collect ground-truth positions and rotations per agent across frames
>         convert positions from relative to absolute reference frame using GPS/IMU
> downsample all data to 2 Hz (smallest common frequency)
> split sequences into 8 historical + 8 future timesteps
> spatially cluster instances so each scene has ~10 agents
> ```
> This shows that creating a unified, structured text representation across heterogeneous datasets is nontrivial. **Similar approaches are adopted in other benchmarks** such as nuScenes-SpatialQA, where annotations from existing datasets are restructured to form new tasks in different domains.
>
> Car4Cast also introduces **LLM-specific metrics** (formatting accuracy, instance precision/recall) which are critical for evaluating structured spatial reasoning in language models. These metrics **address key limitations of current LLMs** in this kind of structured spatial reasoning task and have **direct safety implications**: hallucinated or missing entities, as well as misformatted predictions, could lead to collisions or unsafe decisions in downstream driving systems.
>
> 2) Insights compared to NuScenes-SpatialQA:
>
> NuScenes-SpatialQA evaluates both **qualitative (close-ended)** and **quantitative (numerical)** spatial reasoning. Their findings indicate that **VLMs perform moderately on qualitative questions**, where the nature of token-based cross-entropy loss is still well-suited. However, they show that these models **fail on quantitative tasks requiring numerical predictions, a similar setting to our benchmark**. Even spatially-enhanced VLMs like SpatialRGPT show no clear advantage in this kind of quantitative questions. Furthermore, models with the highest tolerance-based accuracy often have the **largest mean errors**, demonstrating **large errors in some predictions**. **These results align closely with our findings**: most LLMs surpass linear baselines in the median case but underperform in the mean, reflecting large failures in specific scenes.
>
> 3) Proposed solutions for improving performance:
>
> We **suggested potential approaches** in our conclusion, including **reinforcement learning-based fine-tuning** with geometry-aware rewards, or **architectures with activable numeric regression heads**. As a dataset paper, our primary goal is to highlight unexpected outcomes and underexplored weaknesses; **providing full solutions is out of scope**. **Hard benchmarks like Car4Cast are valuable** also because they surface these challenges.
>
> 4) Non-reasoning models outperforming reasoning models:
>
> This highlights a **key difference between quantitative spatial reasoning and traditional logical reasoning tasks**. While surprising, it points to promising research directions for reasoning models. NuScenes-SpatialQA reports similar findings: **introducing chain-of-thought prompting can degrade quantitative spatial reasoning performance**. Our results corroborate that common reasoning paradigms do not necessarily improve performance on spatial reasoning tasks.

---

> > ### Author Response · Authors · 2025-11-21
> >
> > 5) Compatibility with other LLM-based models and evaluation:
> >
> > Car4Cast is **fully compatible** with LLMs: **any model** that outputs predictions in the **same structured JSON format as the inputs** can be evaluated using our provided toolkit. The provided evaluation suite computes all metrics automatically, ensuring reproducibility and facilitating adoption by the community.
> >
> > **TrajLLM** ingests data in the **ChatML format (similar to our finetuning setup)**, hence it is also **compatible with Car4Cast**. While, at train time, TrajLLM expects additional inputs like map layouts and agent behaviors, these modalities are optional at test time and the model can be prompted as a regular LLM. However, since they do not provide the full processed dataset (or scripts to process raw data) nor trained model checkpoints, we were unable to test it directly.
> >
> > 6) Using higher-resolution maps:
> >
> > While RGB top-down views are not available in our source datasets, we value the relevance of investigating the **impact of higher-resolution map inputs**. As a demonstration, we **collected and rendered high-resolution semantic road maps** from the “mini” split of nuScenes and used them as image prompts to VLMs. An example of such map can be found in the nuScenes documentation (https://www.nuscenes.org/nuscenes?tutorial=maps). **We will add all available maps in our final version.**
> >
> > Evaluation on nuScenes-mini shows three settings: **no map, LiDAR top-down map (as in the paper), and semantic map input**.
> >
> > | Model                             | # Params | ACC_f ↑  | ADD (mean) ↓ | ADD (median) ↓ | F1_instance ↑ | P_instance ↑ | R_instance ↑ |
> > |-----------------------------------|----------|----------|--------------|----------------|---------------|--------------|--------------|
> > | Llama-4-Maverick-17B              | 17B      | 0.947    | 2.690        | 0.877          | 0.932         | **0.889**    | 0.979        |
> > | Llama-4-Maverick-17B (LiDAR map) | 17B      | 0.976    | **2.593**    | **0.792**      | **0.938**     | 0.886        | **0.997**    |
> > | Llama-4-Maverick-17B (semantic map) | 17B    | **0.984** | 59.207      | 0.973          | 0.936         | **0.889**    | 0.987        |
> > | | | | | | | | |
> > | gemma-3-27b-it                    | 27B      | 0.997    | 5.907        | 2.980          | 0.803         | 0.802        | 0.803        |
> > | gemma-3-27b-it (LiDAR map)        | 27B      | **1.000** | 5.982       | **2.671**      | 0.831         | 0.829        | 0.833        |
> > | gemma-3-27b-it (semantic map)     | 27B      | **1.000** | **5.665**   | 2.721          | **0.846**     | **0.842**    | **0.851**    |
> > | | | | | | | | |
> > | Qwen2.5-VL-32B                    | 32B      | 0.996    | 75.004       | 1.652          | 0.934         | 0.884        | 0.990        |
> > | Qwen2.5-VL-32B (LiDAR map)        | 32B      | **1.000** | **2.756**   | **0.830**      | 0.935         | 0.887        | 0.988        |
> > | Qwen2.5-VL-32B (semantic map)     | 32B      | **1.000** | 9.448       | 1.154          | **0.938**     | **0.888**    | **0.993**    |
> >
> > We observe that **semantic maps typically do not improve geometric metrics**, even compared to sparse, lower-resolution LiDAR maps. This suggests that **performance limitations** of VLMs with map inputs likely **do not stem solely from map resolution**, but rather from VLMs’ lack of spatial-geometric capabilities of interpreting a map and correctly relating coordinates on axes to their real-world meanings.
> >
> > We hope the above clarifications address all raised concerns. We are happy to provide further details or additional clarifications if needed.

---

### Official Review · Reviewer_jh9F · 2025-11-01

**Soundness:** 2
**Presentation:** 2
**Contribution:** 2
**Rating:** 6
**Confidence:** 4

**Summary:**

This paper proposes Car4Cast, a dataset and benchmark designed for improving LLM-based spatial reasoning in autonomous driving using 3D motion forecasting. Car4Cast uses trajectory prediction and formulates a structured text generation task, requiring LLMs to predict future positions and orientations of vehicles in JSON-like format.

Key contributions of Car4Cast are:
- Dataset: ~11,985 scenes, 102,510 annotated instances from established 3D driving datasets (nuScenes, KITTI, Argoverse, etc.), with both textual and optional visual modalities (LiDAR-based maps).
-  Combines classical metrics (ADE, FDE, rotation error) with LLM-specific metrics (formatting accuracy, hallucination rate, collision rate).
- Experiments: Benchmarks multiple LLMs (14B–1000B parameters) and VLMs, analyzing effects of reasoning, vision modality, and supervised fine-tuning

**Strengths:**

- The proposed dataset addresses a critical gap in evaluating spatial reasoning for LLMs.
- The proposed dataset and evaluation metric includes both traditional forecasting metrics and LLM-specific criteria.
- Spatial reasoning is essential for autonomous driving; the paper convincingly argues why LLMs struggle.
- The paper conducts an insightful experimental analysis which covers reasoning, vision modality, and fine-tuning effects.

**Weaknesses:**

- The proposed Car4Cast has limited real-world deployment perspective, since it mainly focuses on benchmarking using nuScenes, KITTI, Argoverse, etc.. rather than practical integration into driving stacks.
- Can authors explain why  even best LLMs fail to match simple Linear baselines in mean ADD.
- In Table 2 it shows clearly that visual modality is underutilized, why does VLMs perform worse than text-only models; It would be better to see an analysis of this issue, since currently this experiment is shallow not giving any reasoning or perspective.
- In the current experimental set up fine-tuning has limitations:  authors did not explore slternative strategies (e.g., RLHF, geometry-aware loss) experimentally.
- Experiments regarding safety implications are not discussed, i.e authors did not discuss on how hallucinations or formatting errors could impact downstream driving systems using the proposed Car4Cast dataset.

**Questions:**

- Could chain-of-thought reasoning combined with physics priors improve performance on nonlinear trajectories?
- How robust is Car4Cast to domain shifts (e.g., different cities, weather, or sensor setups)?
Please refer weaknesses

---

> ### Author Response · Authors · 2025-11-21
>
> Thank you for your time and feedback. Below we address the raised weaknesses and questions. Please don’t hesitate to ask if anything remains insufficiently addressed.
>
> 1) Practical integration into driving stacks:
>
> We fully agree that ultimately we would like to see strong AI models integrated in autonomous driving scenarios. However, as the Car4Cast results show, **current LLMs are so weak at understanding motion that it would be ill-advised to use them in such applications**. Car4Cast helps **narrow this capability gap** by providing a **testbed for improvements** in LLM spatial reasoning and motion understanding.
>
> 2) Failure to match simple baseline:
>
> Given that the **median ADD is much better than the average, robustness appears to be a key issue**. In particular, we observe that the mean ADD is **strongly driven by catastrophic failures on some scenes**. For instance, Qwen3-235B-A22B in non-reasoning mode shows over 4000 ADD on scene `ea6895f2-504b-37b5-bfd0-cbf7017f22c3_1_1`, where all instances are predicted to be stationary at some location which is notably far from any of the historical or future trajectories. A **visualization** of historical, ground-truth future, and predicted future trajectories is available in our **updated supplementary material** at `visual_analysis/ea6895f2-504b-37b5-bfd0-cbf7017f22c3_1_1_Qwen3-235B-A22B.mp4`
>
> 3) Why do VLMs perform worse than text-only models?
>
> As a dataset paper, a key goal is to **highlight unexpected outcomes and currently underappreciated weaknesses**; in this sense, the discovery of this behavior is a benefit of our dataset. Additional results show that **other visual data (namely point-of-view images from the recording car) can improve outputs**, suggesting that mapping from top-down views to motion is part of the challenge. Below we present the results of VLMs evaluated on scenes extracted from the nuScenes-mini set, comparing their performance with or without the front-facing camera input.
>
> | Model                           | # Params | ACC_f ↑  | ADD (mean) ↓ | ADD (median) ↓ | F1_instance ↑ | P_instance ↑ | R_instance ↑ |
> |---------------------------------|----------|----------|--------------|----------------|---------------|--------------|--------------|
> | Llama-4-Maverick-17B            | 17B      | 0.947    | 2.690        | 0.877          | 0.932         | 0.889        | 0.979        |
> | Llama-4-Maverick-17B (with camera) | 17B   | **0.994** | **2.436**   | **0.769**      | **0.939**     | **0.895**    | **0.987**    |
> | | | | | | | | |
> | gemma-3-27b-it                  | 27B      | 0.997    | 5.907        | 2.980          | 0.803         | 0.802        | **0.803**    |
> | gemma-3-27b-it (with camera)    | 27B      | **1.000** | **5.788**   | **2.429**      | **0.817**     | **0.845**    | 0.791        |
> | | | | | | | | |
> | Qwen2.5-VL-32B                  | 32B      | **0.996** | 75.004      | 1.652          | **0.934**     | 0.884        | **0.990**    |
> | Qwen2.5-VL-32B (with camera)    | 32B      | 0.936    | **4.131**    | **1.596**      | 0.931         | **0.892**    | 0.972        |
>
> 4) Why no RLHF evaluation or geometry-aware losses?
>
> As discussed in the paper, this is a promising avenue for future work. The goal of the present dataset paper is to highlight such directions; **following through on them is out of scope for this work**.

---

> > ### Author Response · Authors · 2025-11-24
> >
> > 5) Safety implications:
> >
> > We agree that the safety implications of spatial reasoning and LLM outputs are an important aspect. **We will add the following discussion to the paper.**
> >
> > **Spatial reasoning is critical for safety in autonomous driving**. Understanding how objects relate to each other and their positions in the environment directly affects perception, planning, and decision-making, including navigation, collision avoidance, and interaction with other vehicles. Some common error scenarios involving language models include:
> > - **Hallucinated entities**: Predicting a car at a location where none exists could cause unnecessary **sudden braking**.
> > - **Missing entities**: Failing to predict a car’s trajectory could make the ego vehicle **incorrectly assume a safe path**, risking collisions.
> > - **Formatting errors**: Forecasted positions may be **unreadable**, leading to **ignored objects** and potential accidents.
> >
> > 6) Evaluation of physics-informed chain-of-thought?
> >
> > Similar to point 4, this is an **interesting avenue for future work**. The purpose of this dataset paper is to **enable exploration of such directions**. **Evaluating them is out of scope**; we do show that standard approaches like prompt engineering and fine-tuning fail. **Machine learning is a benchmark-driven field** (consider, e.g., ImageNet -> ResNets); our goal is to nudge the field toward the directions you suggest.
> >
> > 7) Robustness to domain shift:
> >
> > Car4Cast is composed of **multiple source datasets**, allowing for **domain shift experiments**: the datasets include **different sensor setups, cities, and weather conditions**. We plan to **add labels to distinguish sensor setups** and **point-of-view images** to enable weather filtering.
> >
> > Thank you again for your helpful input. We believe these additions and clarifications will improve our paper. Please do not hesitate to provide further suggestions, questions, or comments.

---

### Author Response · Authors · 2025-12-03
**Rebuttal Summary for the Area Chair**

We thank the reviewers for their time and for the constructive feedback. Due to the premature termination of the reviewers' feedback window, only one reviewer was able to respond during the discussion. That reviewer explicitly **raised their score from 6 to 10** on November 25 (as they previously anticipated in the original review), noting that our rebuttal fully resolved their concerns. The remaining reviewers did not have time to reply, however, we still provided detailed answers and substantial new material **addressing all questions raised.**

### Summary of contributions during the discussion period

### **1. Deeper analysis of the visual modality for VLMs (new modalities + new experiments)**
To deepen our analysis on VLMs and better answer reviewer concerns, we added **two new visual modalities** beyond the LiDAR BEV maps included in the paper:
- **Front-facing RGB camera images**
- **High-resolution semantic top-down maps**

These new experiments led to the following observations:
- **Semantic maps *do not* improve geometric accuracy**, even compared to sparse, noisy, LiDAR maps. This suggests that limitations of VLMs with map inputs are not due to map resolution, but due to limited spatial-geometric capabilities (e.g., interpreting axes, associating coordinates with real-world meaning).
- **Camera inputs *do* improve distance-based metrics**, suggesting that VLMs have the potential to extract meaningful information from the RGB modality

### **2. Stronger and more informative motion forecasting baselines (new experiments)**
Responding to requests for comparisons with more conventional motion forecasting methods, we added **two additional baselines** on top of our linear predictor introduced in the paper:
- A **two-layer MLP** predicting residuals relative to a linear extrapolation
- A **12-D constant-velocity Kalman Filter**

**Both baselines outperform the simple linear model (and therefore all the tested LLMs/VLMs)** when averaged over the full test set.
This:
- Provides the community with harder, more realistic baselines
- Reinforces our key empirical finding: **LLMs/VLMs currently underperform even a simple linear forecast, so they also fall short of more specialized methods.** This gap is exactly the weakness Car4Cast is designed to quantify and narrow

### **3. Added qualitative video visualizations (new supplementary material)**
We added **video visualizations** (historical trajectories, predictions, and ground-truth) for scenes across different model families, to provide **visual examples of some experimental observations we made in the paper**:
- Catastrophic LLM failures driving mean ADD
- Cases where reasoning leads to better linear-motion extrapolations
- VLMs improving formatting but degrading geometrical accuracy
- Supervised Fine-Tuning improving JSON formatting while worsening spatial accuracy

### **4. Clarified benchmark compatibility and ease of adoption**
We expanded our explanation of how models can be evaluated on our benchmark:
- **Any model can be evaluated** by outputting predictions in the same structured JSON format as the inputs
- The **released evaluation code** automatically computes all metrics
- This ensures full reproducibility and simplifies community adoption (addressing the concern about evaluation standardization)

### **5. Clarified the scope of our dataset/benchmark paper**
We emphasized that Car4Cast’s purpose, as a dataset and benchmark paper, is to:
- **reveal spatial reasoning weaknesses** of LLMs/VLMs
- highlight unexpected model behaviors
- provide a **standardized playground for future research development and evaluation**

We additionally **proposed future directions** to address the exposed gap (e.g., RLHF with geometry-aware rewards or architectures with numeric regression heads), clarifying that they are are outside the scope of this dataset paper.

**Benchmarks typically play this role in machine learning**: they expose failure modes and motivate the next generation of methods.

---

We believe our additions substantially strengthen the paper and fully address reviewers’ concerns.

---

### Meta-Review · Area_Chair_HvQG · 2026-01-07

**Summary:**

While we recognize the substantial engineering effort involved in creating Car4Cast and the value of unifying multiple driving datasets, the decision is to reject the paper at this time. The consensus among the majority of reviewers is that the paper falls short of the acceptance bar due to significant concerns regarding the practical utility of the proposed benchmark and the limited methodological novelty. Specifically, the primary "fact" established by the study—that Large Language Models (LLMs) and Vision-Language Models (VLMs) are significantly outperformed by trivial baselines like a linear predictor or a 2-layer MLP—suggests that this formulation may not yet be a viable testbed for driving progress in motion forecasting. Furthermore, the premise of framing this as a "spatial reasoning" task is weakened by findings that explicit reasoning often degrades performance and that visual inputs (maps) fail to improve geometric accuracy. Consequently, the work is viewed currently as a robust engineering compilation rather than a benchmark that offers the novel insights or significant results required for this conference.

**Reviewer Concerns:**

Addressed by Rebuttal:

    ◦ Baselines: The reviewers appreciated the authors' responsiveness in adding requested baselines, including a 2-layer MLP and a Kalman Filter.
    ◦ Modalities: The inclusion of semantic top-down maps and front-facing camera images addressed the initial concern regarding missing visual inputs.
    ◦ Reproducibility: The authors successfully clarified their evaluation pipeline and committed to releasing code to aid community adoption.

Outstanding:

    ◦ Performance Gap & Utility: A critical outstanding concern is the "capability deficiency" exposed by the new baselines. The fact that a simple 2-layer MLP (ACC_f 1.528) and a Kalman Filter (ACC_f 1.194) outperform all tested LLMs (e.g., Qwen3 non-reasoning at 1.792) suggests that LLMs are currently "ill-advised" for this specific formulation of the task. This raises doubts about the benchmark's immediate utility for fostering modeling improvements.

    ◦ Novelty vs. Engineering: Reviewers maintained that the contribution remains primarily data processing—extracting and reformatting existing datasets (nuScenes, KITTI, etc.)—rather than offering a novel scientific methodology.

    ◦ Ineffectiveness of Reasoning & Vision: The rebuttal confirmed that "reasoning" models often perform worse than non-reasoning ones, and adding visual maps frequently resulted in lower performance on geometric metrics. These findings undermine the paper's core motivation to use this task as a proxy for evaluating spatial reasoning or multimodal integration.

**Reviewer Scores:**

• Reviewer H1Lv (Final Score: 10): This reviewer was highly encouraged by the rebuttal and the addition of baselines, viewing the dataset as "extremely valuable to the community" regardless of the model performance. However, this enthusiastic view was an outlier compared to the concerns regarding scientific soundness raised by others. Yet, this revised score happens after the data leakage.

• Reviewer Fw34 (Initial: 2): This reviewer remained concerned that the paper does not sufficiently justify why motion forecasting should be framed as a spatial reasoning task for LLMs, especially given that traditional approaches work significantly better. The rebuttal results likely confirmed their skepticism regarding the validity of the proposed direction.

• Reviewer fbaA (Initial: 4): This reviewer focused on the lack of novelty, noting the contribution was "merely the combination of datasets". While they found the negative results informative, the consistent failure of LLMs against linear baselines likely prevented them from raising their score to an acceptance level.

• Reviewer jh9F (Initial: 6): While acknowledging the dataset fills a gap, this reviewer raised concerns about why VLMs perform worse than text-only models. The rebuttal confirmed this counter-intuitive finding without fully resolving why it occurs, which likely limited their enthusiasm for the paper's final contribution.

---

### Decision · Program_Chairs · 2026-01-26

Reject